# Hippocampal theta coordinates memory processing during visual exploration

James E Kragel[1]*, Stephen VanHaerents[2], Jessica W Templer[2], Stephan Schuele[2], Joshua M Rosenow[3], Aneesha S Nilakantan[1], Donna J Bridge[1]*

[1]Department of Medical Social Sciences, Northwestern University Feinberg School of Medicine, Chicago, United States; [2]Department of Neurology, Northwestern University Feinberg School of Medicine, Chicago, United States; [3]Department of Neurological Surgery, Northwestern University Feinberg School of Medicine, Chicago, United States

**Abstract** The hippocampus supports memory encoding and retrieval, which may occur at distinct phases of the theta cycle. These processes dynamically interact over rapid timescales, especially when sensory information conflicts with memory. The ability to link hippocampal dynamics to memory-guided behaviors has been limited by experiments that lack the temporal resolution to segregate encoding and retrieval. Here, we simultaneously tracked eye movements and hippocampal field potentials while neurosurgical patients performed a spatial memory task. Phase-locking at the peak of theta preceded fixations to retrieved locations, indicating that the hippocampus coordinates memory-guided eye movements. In contrast, phase-locking at the trough of theta followed fixations to novel object-locations and predicted intact memory of the original location. Theta-gamma phase amplitude coupling increased during fixations to conflicting visual content, but predicted memory updating. Hippocampal theta thus supports learning through two interleaved processes: strengthening encoding of novel information and guiding exploration based on prior experience.

*For correspondence:
james.kragel@northwestern.edu
(JEK);
donnajb@gmail.com (DJB)

**Competing interests:** The authors declare that no competing interests exist.

## Introduction

Hippocampal theta rhythms are prominent during active exploration of novel environments, perhaps due to encoding and retrieval processes necessary to guide ongoing behavior. Interactions between encoding and retrieval support many important functions, including memory updating, which requires comparing novel sensory inputs to prior memories and integrating new content into memory representations (*Bridge and Paller, 2012*; *Bridge and Voss, 2014b*). Retrieval-mediated reconsolidation requires the presence of novel information during reactivation, suggesting that this hippocampal-dependent learning process is sensitive to mismatch (associative novelty) between the retrieved content and sensory input (*Morris et al., 2006*; *Winters et al., 2011*). Some studies have even demonstrated hippocampal involvement in associative novelty (*Bridge and Voss, 2014a*; *Chen et al., 2013*; *Duncan et al., 2009*; *Duncan et al., 2012*; *Honey et al., 1998*; *Howard et al., 2011*; *Kumaran and Maguire, 2007a*; *Kumaran and Maguire, 2009*; *Long et al., 2016*; *Thakral et al., 2015*). However, the underlying novelty and retrieval processes have not been segregated as they unfold, in part because it is difficult to segregate these mechanisms in real time, as they interact continuously during learning. Many experimental designs capitalize on an artificial separation between encoding and retrieval phases, but these designs do not capture the natural interplay between these states that guides exploratory behavior and informs decision-making. Here, we assayed the engagement of encoding and retrieval processing in real time by designing a task to link memory-guided eye movements to intracranial recordings of hippocampal activity, and aimed to identify how theta oscillations are distinctly involved in encoding and retrieval processes.

Eye movements provide rich temporal information regarding the focus of attention and the specific cognitive processes engaged at any given moment (*Bridge et al., 2017*; *Bridge and Voss, 2014b*; *Bridge and Voss, 2015*; *Voss et al., 2011*). In human and non-human primates, learning through exploration heavily depends on the visual system (*Meister and Buffalo, 2016*), with eye movements resetting the phase of theta during learning of novel visual information (*Hoffman et al., 2013*; *Jutras et al., 2013*). Eye movements are also deployed rapidly, with median saccade rates of 3–5 Hz during visual exploration tasks (*Wilming et al., 2017*). Thus, eye movements provide an ideal behavioral measure to dissect learning processes on a moment-to-moment basis.

The interaction of fast gamma and theta oscillations in the hippocampus could play a key role in coordinating interactions between encoding and retrieval during exploration. The amplitude of gamma increases at specific phases of the theta cycle to support memory processing (*Axmacher et al., 2010*; *Lega et al., 2016*). Computational models of hippocampal function suggest that gamma activity associated with encoding and retrieval preferentially occurs at the trough and peak of the theta rhythm, respectively (*Hasselmo et al., 2002*). Task-based observations of hippocampal firing support this idea, showing that novel stimulus encoding and memory retrieval are enhanced at distinct phases of the theta cycle (*Douchamps et al., 2013*; *Lever et al., 2010*; *Manns et al., 2007*; *Newman et al., 2013*). In addition, closed-loop optogenetic stimulation of inhibitory neurons aligned to the peak of theta improves encoding, whereas stimulation aligned to the trough improves retrieval (*Siegle and Wilson, 2014*).

In rodents, theta-modulated gamma band activity has also distinguished encoding from retrieval (*Bragin et al., 1995*; *Colgin, 2015*). Distinct slow (~30 to 50 Hz) and mid (~50 Hz to 100 Hz) gamma oscillations are observed and generated by separate neural circuits, with maximal amplitudes at the peak and trough of the theta rhythm, respectively (*Colgin, 2016*). Recent work has also identified unique theta-nested gamma oscillations observed within individual theta cycles (*Lopes-Dos-Santos et al., 2018*), providing support for the notion that fast and slow gamma separately mediate encoding of novel information and memory retrieval. Intracranial recordings in humans have identified ripple oscillations (~80 to 100 Hz) that are involved in memory retrieval and consolidation (*Axmacher et al., 2008*; *Staresina et al., 2015*; *Vaz et al., 2019*) and exhibit phase amplitude coupling (PAC) with hippocampal theta phase (*Staresina et al., 2015*). Similar ripple oscillations are also prevalent in nonhuman primates during visual search (*Leonard and Hoffman, 2017*; *Leonard et al., 2015*), raising the possibility that they play an active role in exploration. Because evidence for memory-related hippocampal theta to gamma PAC in humans has primarily focused on verbal learning tasks (*Lega et al., 2016*; *Mormann et al., 2005*), it is not known how changes in PAC during exploration translate from animal models to similar behaviors in humans.

Here, we recorded eye movements and intracranial hippocampal recordings as neurosurgical patients performed an associative spatial memory task. We hypothesized that theta oscillations would influence when eye movements were driven by either associative novelty or memory retrieval, through theta-dependent modulation of neuronal firing. Indeed, previous work in humans has examined the relation between theta oscillations and memory in verbal recall tasks (*Kahana et al., 2001*; *Lega et al., 2012*; *Sederberg et al., 2003*), including theta-gamma phase amplitude coupling (*Lega et al., 2016*; *Mormann et al., 2005*; *Vaz et al., 2017*). These studies examined encoding and retrieval in isolated task epochs, raising the question of how theta supports these memory processes when they rapidly co-occur. In our spatial memory task, subjects encountered previously studied objects in either their original or updated spatial locations. When an object appeared in an updated location, subjects directed viewing to both the updated and original locations iteratively over the course of the trial. Simultaneous acquisition of eye movements and intracranial EEG allowed us to relate theta to the timing of encoding and retrieval processes with exceptional temporal resolution. In doing so, we systematically tested the hypothesis that hippocampal theta influences when different memory processes occur during exploratory viewing. If the strength of encoding and retrieval are modulated by theta phase, fixations driven by each process should be phase-locked to the theta rhythm. In addition, eye-movements tied to distinct encoding and retrieval processes should occur at distinct phases of theta with variation in hippocampal PAC. As such, this experiment determines how the hippocampus contributes to learning and coordinates dynamic encoding and retrieval operations during visual exploration in humans.

## Results

### Direct brain recordings linked to memory-guided eye movements

Subjects performed a multi-phase associative spatial memory task (*Figure 1a*), while we simultaneously recorded eye movements and local field potentials from the hippocampus (*Figure 1b*). During the study phase, subjects learned the spatial location of 16 objects presented sequentially on a background scene. Next, during a refresh phase, objects were re-presented in either repeated (Match) or updated (Mismatch) spatial locations, with two visual cues (small red dots) indicating potential alternate locations. One visual cue always indicated the object's original location during Mismatch trials. After viewing each stimulus, subjects indicated via button press whether each object was presented in its original or updated location. During a final recognition phase, subjects viewed each object in three locations and attempted to identify the object's original location. All subjects completed eight blocks of the study-refresh-recognition sequence, with 16 unique objects per block (128 total), and a unique background scene per block (eight total).

During the refresh phase, subjects were encouraged to visually explore the three cued locations to help inform their memory decision. Our primary analyses focused on the interplay of associative novelty and retrieval processes during these Mismatch trials, by linking hippocampal activity to eye movements directed to the original and updated locations. We use the term retrieval-dependent to refer to fixations to the original object-location on Mismatch trials, as retrieval of spatial information is necessary for preferential viewing of this location. Novelty-dependent fixations are driven to updated object-locations, as compared to fixations to objects presented in repeated locations during Match trials. By leveraging eye movements in this manner, we were able to identify distinct hippocampal mechanisms linked to these cognitive processes. In addition, we evaluated the impact of viewing behaviors and electrophysiological states on final recognition performance.

We measured overall task performance by computing accuracy on the final recognition test. Subjects performed the task well, correctly identifying repeated object-locations on 72% (±6 SEM) of Match trials and novel object-locations on 70% (±6 SEM) of Mismatch trials during the refresh phase. Subjects remembered the original object-location on 72% (±6 SEM) of final recognition test trials (*Figure 1c*). We assessed how the factors of condition (Mismatch/Match) and refresh task performance (Correct/Incorrect) influenced memory for the original object-location using a two-factor repeated measures ANOVA. We observed significant main effects of refresh performance ($F_{1,4}$ = 115.0, p = 0.0004, $\eta_p^2$ = 0.97) and condition ($F_{1,4}$ = 13.4, p = 0.02, $\eta_p^2$ = 0.77), without evidence for an interaction ($F_{1,4}$ = 0.4, p = 0.55, $\eta_p^2$ = 0.10). Recognition accuracy on the final test was significantly better following correct (M = 84, SD = 10) than incorrect (M = 35, SD = 15) judgments on the refresh phase (paired t-test, $t_4$ = 10.72, p = 0.0004, g = 3.4). In addition, accuracy on the final recognition test was significantly impaired on Mismatch (M = 63, SD = 14) relative to Match trials (M = 82, SD = 14; paired t-test, $t_4$ = 9.83, p = 0.0006, g = 1.2).

To confirm that eye movements during the refresh phase were tied to memory processes, we examined changes in viewing behaviors and memory outcomes on the final recognition test (*Table 1*). On average, participants made more fixations to the presented object during Match (M = 5.3, SD = 1.0; paired t-test, $t_4$ = 12.9, p = 0.0002, g = 2.9) and Mismatch (M = 4.6, SD = 0.9; paired t-test, $t_4$ = 17.7, p = 0.0001, g = 2.8) trials than to the other two cued locations on Mismatch (M = 2.4, SD = 0.8) and Match (M = 1.7, SD = 0.8) trials. Notably, the number of fixations to the object was reduced on Mismatch relative to Match trials (paired t-test, $t_4$ = −8.8, p = 0.0009, g = −0.6), indicating increased exploration during Mismatch. In addition, fixation durations to objects were longer than matched spatial cues (paired t-test, all $t_4$ > 3.4, p < 0.03, g > 1.8) and were comparable between Match (M = 342 ms, SD = 88) and Mismatch trials (M = 388, SD = 131; paired t-test, $t_4$ = 1.49, p = 0.21, g = 0.4).

Viewing behavior on Match and Mismatch trials predicted final recognition performance. On Match trials, the number of fixations to the repeated object predicted better memory for the original location, whereas the number of fixations to the updated location on Mismatch trials predicted memory updating (see *Table 1* for details). To break down the timing of these memory-guided eye movements, we examined the proportion of time spent viewing each region of interest (ROI) across trials (*Figure 1d*). We found that viewing preferences on Mismatch trials, but not Match trials, predicted later memory for the original object-locations. Following initial visual orienting to the novel stimulus, prolonged viewing of the novel object-location (738 to 2188 ms after object presentation)

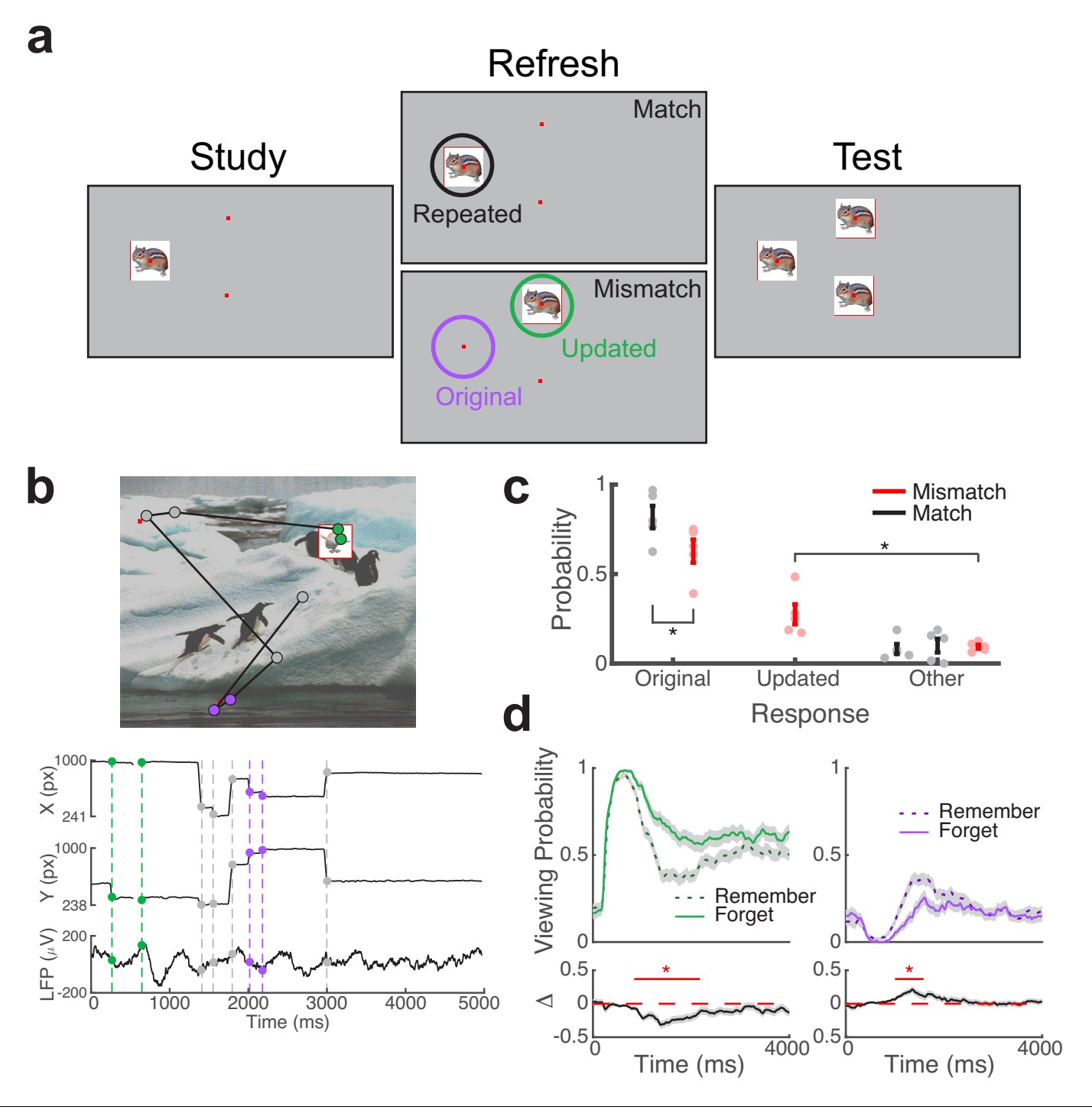

**Figure 1.** Direct brain recordings during memory-driven eye movements. (a) Spatial memory task. Example stimuli presented during each phase of the task. Viewing regions of interest (ROIs) for each trial type are indicated by circles on the Refresh phase. (b) Simultaneous recording of gaze position and hippocampal field potential during an example trial. Above, viewing scan path overlaid on the stimulus display for a Mismatch trial. Below, gaze position and concurrent signal for an electrode in the hippocampus. The onset of fixations to viewing ROIs are denoted by colored circles. (c) Behavioral performance. Response proportions on the final recognition test for each viewing condition. Each point denotes a subject average; lines denote one SEM. (d) Viewing behavior on Mismatch trials predicts memory outcomes. The probability of viewing the updated (left) or original (right) object-location was compared on trials in which the original location was subsequently remembered or forgotten. Below, a subsequent memory effect was computed as the difference in viewing probability. Shaded areas depict ± SEM. Lines depict significant clusters ($P_{FWE} < 0.05$).

The online version of this article includes the following source data for figure 1:

**Source data 1.** MATLAB code and source files to reproduce data in *Figure 1*.

**Table 1.** Task-related eye movement behavior.

| | Mismatch | | | Match | | |
|---|---|---|---|---|---|---|
| | Original | Updated | Other | Repeated | Other | Other |
| Fixations per trial (N) | 2.4 (0.3) | 4.6 (0.4) | 1.4 (0.3) | 5.3 (0.4) | 1.7 (0.4) | 1.4 (0.3) |
| Fixation SME (*t*) | 1.4 | −4.6[*] | −0.3 | 3.7[*] | −0.4 | −1.7 |
| Fixation duration (ms) | 210 (11) | 388 (59) | 212 (14) | 342 (39) | 199 (11) | 206 (6) |
| Duration SME (*t*) | 0.5 | 1.1 | 1.4 | −1.6 | −2.4 | −0.6 |

Group-level description of eye movement behavior to six viewing regions of interest. The subsequent memory effect (SME) for each measure was assessed by a one-sample *t*-test, across subjects (n = 5). [*]$P < 0.05$. Parentheses denote standard error of the mean.

was associated with memory updating (nonparametric cluster test, $P_{FWE} < 0.05$, g = −0.7, n = 5 subjects). Viewing the original object-location during this time period (from 998 to 1584 ms) led to better memory for the original location (nonparametric cluster test, $P_{FWE} < 0.05$, g = 1.2, n = 5 subjects). Proportion of viewing over time during Match trials was not a significant predictor of final memory performance ($P_{FWE} > 0.05$). These findings suggest that interplay between memory processes and visual sampling during Mismatch trials determined whether memory updating would occur.

### Theta dependence of memory-guided eye movements

We analyzed direct recordings from hippocampal depth electrodes in five subjects (*Figure 2a*), referencing signals from contacts in hippocampus or adjacent white matter to bipolar pairs (hereafter called electrodes). Average power spectra from pre- and post-fixation intervals contained theta peaks irrespective of task condition and fixation target (*Figure 2b*). To determine the consistency of these peaks at the individual electrode level (see *Figure 2—figure supplement 1* for individual power spectra), we modeled aperiodic and oscillatory components of power spectra for individual fixations. We tested whether theta (4 to 6 Hz) oscillations were present by comparing variability in the full spectra (which includes oscillatory components) to the amount explained by aperiodic components alone. A majority of electrodes exhibited theta oscillations around this peak frequency both before (−750 to 50 ms) and after (−50 to 750 ms) fixation onsets to each ROI (see *Table 2* for details). In the following analyses, we focused on spectral power and phase from 1 Hz to 10 Hz, which includes both theta and low-theta (*Jacobs, 2014*; *Watrous et al., 2013a*) frequency bands previously associated with visual exploration and memory encoding (*Jutras et al., 2013*; *Lega et al., 2012*). We linked measures of spectral power and phase to individual fixation events to identify hippocampal states reflecting retrieval and novelty detection. We reasoned that hippocampal signaling prior to fixations would reflect a memory-guided initiation of the upcoming eye movement, whereas signaling following fixations would reflect a memory-based reaction to visual input.

### Theta phase coherence increases during retrieval and novelty detection

To assess whether retrieval-dependent eye movements during Mismatch trials occurred at specific phases of hippocampal oscillations, we compared the consistency of phase angles in the moments leading up to fixations to the original and updated locations using inter-trial phase coherence (ITC; *Figure 2c*). We observed significantly greater phase-locking around 5 Hz prior to fixations to the original compared to the updated object-location on Mismatch trials (nonparametric cluster test, $P_{FWE} = 0.045$, g = 1.2, n = 5 subjects). In an additional analysis, we also identified significantly greater phase-locking preceding fixations to the original object-location relative to fixations to objects presented in repeated locations (Match condition) during the same time-frequency window (*Figure 2—figure supplement 2*, $P_{FWE} = 0.04$, g = 3.4, n = 5 subjects). Theta power was comparable before fixations to the original and updated object-locations (one sample t-test, $t_4 = −1.6$, p = 0.17, *Figure 2—figure supplement 3*), indicating that differences in ITC did not result from reductions in theta magnitude.

As fixations to the original object-location were frequently preceded by novelty-driven fixations (*Figure 1d*), it is possible that the observed phase-locking effect resulted from novelty detection rather than retrieval. Two control analyses suggested this was not the case. First, ITC measured

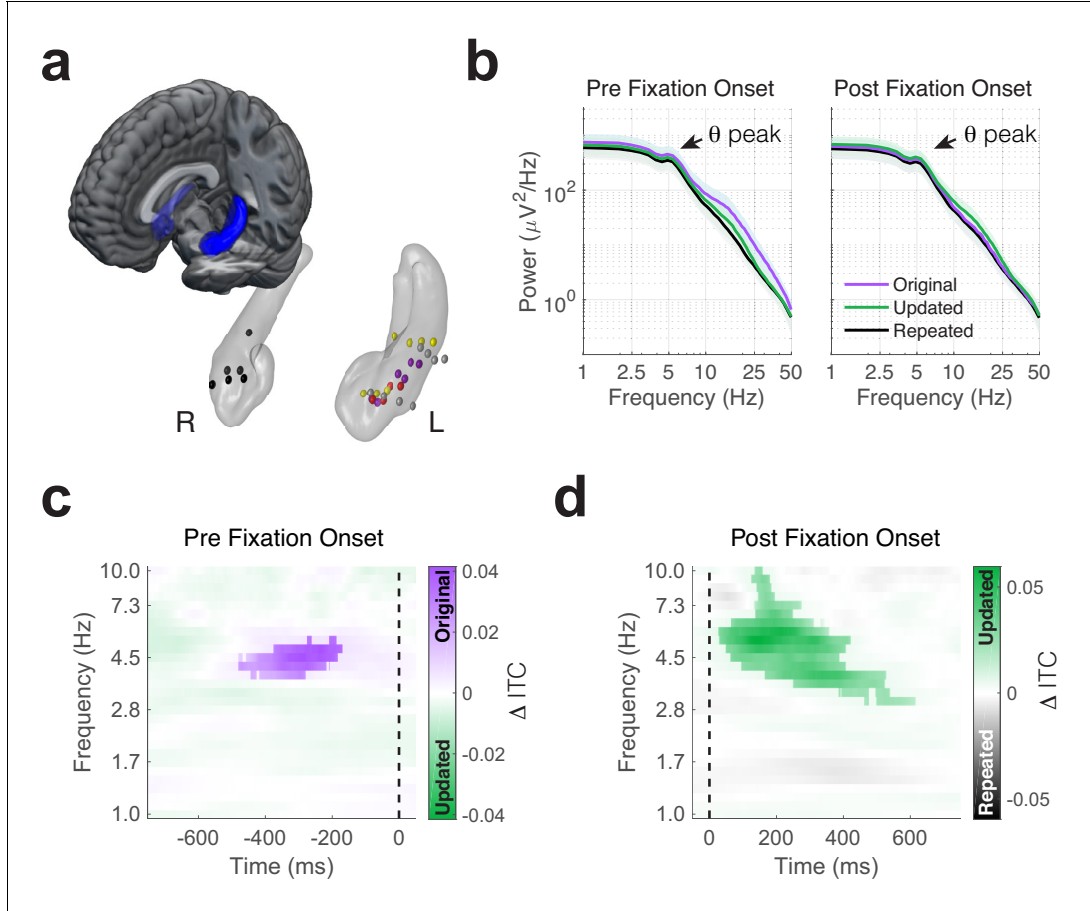

**Figure 2.** Phase-locking of memory-dependent eye movements to hippocampal theta. (a) Location of hippocampal electrodes in MNI space. (b) Mean power spectra in peri-fixation epochs. Spectral peaks in the 4 to 6 Hz range are shown in epochs preceding and following fixations to all regions of interest. (c) Increased phase-locking precedes retrieval-dependent fixations. Significant differences (cluster $P_{FWE} < 0.05$) in inter-trial phase clustering (ITC) between fixations (indicated by the dashed line) to original vs. updated object-locations on Mismatch trials are highlighted. (d) Novelty related modulations in hippocampal phase. Significant differences in ITC following fixations to the updated object-location on Mismatch trials and the repeated object-location on Match trials.

The online version of this article includes the following source data and figure supplement(s) for figure 2:

**Source data 1.** MATLAB code and source files to reproduce data in *Figure 2*.
**Figure supplement 1.** Theta oscillations at individual hippocampal electrodes.
**Figure supplement 1—source data 1.** MATLAB code and source files to reproduce data in *Figure 2—figure supplement 1*.
**Figure supplement 2.** Comparison of inter-trial phase coherence (ITC) between fixations to repeated and original object-locations.
**Figure supplement 2—source data 1.** MATLAB code and source files to reproduce data in *Figure 2—figure supplement 2*.
**Figure supplement 3.** Eye-movement related changes in theta power.
**Figure supplement 3—source data 1.** MATLAB code and source files to reproduce data in *Figure 2—figure supplement 3*.

during fixations to the updated location did not differ depending on the target of the next saccade (i.e. either to the original or updated location; all $P_{FWE} > 0.15$, n = 5 subjects). Second, we observed significantly increased theta phase-locking across subjects (one sample t-test, $t_4 = 2.95$, p = 0.04) during the same time interval when we excluded fixations that were preceded by fixations to the updated object-location (which could cause stimulus-related processing to occur prior to the retrieval-guided fixation). These findings suggest that the observed pre-fixation theta effects reflect a retrieval mechanism, rather than novelty-related processes that initiate memory retrieval.

We next asked whether theta phase was modulated following fixation onset. If theta phase is generally modulated by fixations during memory updating (i.e. during Mismatch trials), consistent phase-locking would occur irrespective of the viewing location. To test this possibility, we contrasted ITC between each type of fixation (to the original or updated location) on Mismatch trials with

**Table 2.** Proportion of electrodes showing theta (4 to 6 Hz) oscillations.

| | N | Pre fixation | | | Post fixation | | |
|---|---|---|---|---|---|---|---|
| | | Original | Updated | Repeated | Original | Updated | Repeated |
| S1 | 8 | 0.75* | 0.75* | 0.38* | 0.25 | 0.5* | 0.5* |
| S2 | 4 | 0 | 0.25 | 0.75* | 0 | 0.5 | 0.75* |
| S3 | 6 | 1.0* | 1.0* | 1.0* | 0.67* | 1.0* | 1.0* |
| S4 | 8 | 1.0* | 1.0* | 1.0* | 1.0* | 1.0* | 1.0* |
| S5 | 6 | 0.83* | 0.83* | 0.83* | 0.5* | 0.67* | 0.67* |
| Group | 32 | 0.78* | 0.81* | 0.78* | 0.53* | 0.75* | 0.78* |

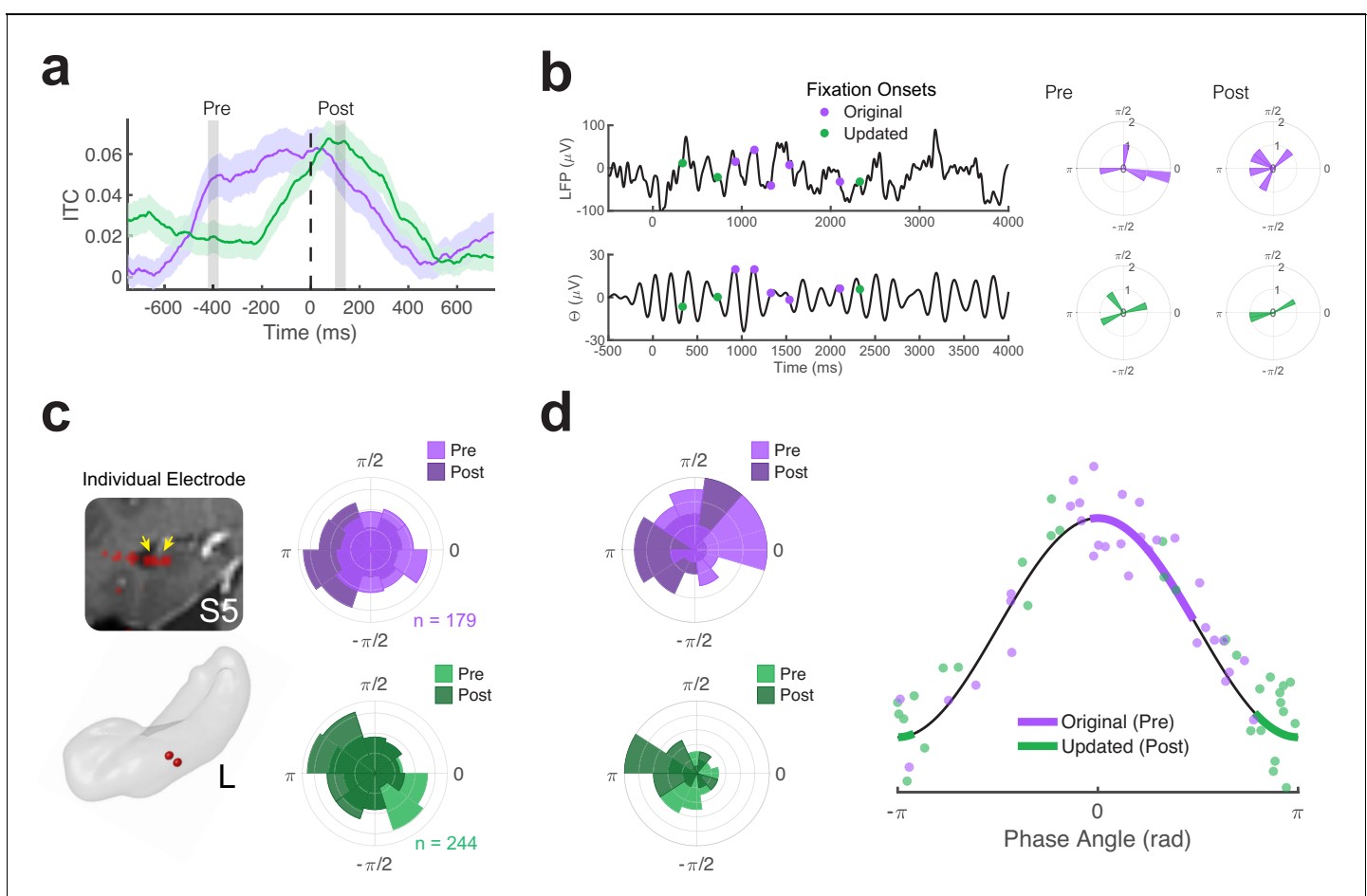

**Figure 3.** Distinct phases of theta phase are associated with retrieval and associative novelty. (a) Timecourse of inter-trial coherence (ITC) for theta (5 Hz) phase during Mismatch trials. Pre- and post-fixation time-periods of interest are indicated by vertical bars. Shaded regions depict ± SEM. (b) Theta phase distributions for an example Mismatch trial. Left, the local field potential measured across an electrode (low-pass filtered at 20 Hz for display) is plotted above the prominent theta (4 to 6 Hz bandpass filtered) timeseries. Time zero denotes the start of the trial. Right, polar histograms show the corresponding distributions of phase angles in the Pre and Post fixation periods for fixations to the original (top panels) and updated (bottom panels) object-locations. (c) Theta phase distributions for a left (L) hippocampal electrode from subject 5 (S5) . Histograms show phase distributions during each time period of interest, aggregated across all fixations of interest during Mismatch trials. (d) Differences in theta phase distributions across all electrodes. Left, polar histograms show the distribution of theta phases across electrodes, averaged for each fixation target. Right, retrieval and novelty-related clustering are associated with distinct phases of theta. Dots depict the average phase angle for each electrode, with 95% confidence intervals indicated by heavy lines.

The online version of this article includes the following source data for figure 3:

**Source data 1.** MATLAB code and source files to reproduce data in *Figure 3*.

fixations to the repeated location on Match trials. We restricted this analysis to trials where memory for the original object-location was intact, to increase the likelihood that fixations were driven by associative novelty rather than purely stimulus-driven factors. Significantly (nonparametric cluster test, $P_{FWE} < 0.001$, $g = 1.2$, n = 5 subjects) greater phase clustering followed fixations to updated compared to repeated object-locations (*Figure 2d*). This post-saccade phase-locking effect only followed fixations to updated locations, as ITC did not significantly differ between fixations to the Mismatch-original and Match-repeated locations. In addition, we did not find any significant differences between phase-locking following fixations to original and updated locations during Mismatch trials. Taken together, our comparison of ITC between fixation targets suggests modulation of theta phase by two processes: retrieval processing preceding fixations to the original location and novelty processing following fixations to the updated location.

## Theta phase angle differentiates retrieval and novelty detection

Models of memory-related oscillations propose that encoding and retrieval operations preferentially occur during the trough ($\pi$ rad) or peak (0 rad) of hippocampal theta, respectively (*Hasselmo et al., 2002*). If theta phase-locking preceding fixations to original object-locations was driven by theta-dependent retrieval mechanisms, we would predict preferred phase angles near the peak of the theta. On the other hand, we would predict the phase-locking to begin at the trough of theta oscillations during fixations to updated object-locations, as theta troughs are associated with increased sensory inputs from the entorhinal cortex that support encoding of the environment.

To test these predictions, we examined differences in theta phase when encoding and retrieval processing occurred on Mismatch trials. The time course of theta coherence (*Figure 3a*) reveals the timing of hippocampal retrieval and novelty processes; a pre-fixation retrieval effect and a reset of theta phase due to fixations to updated object-locations. We compared phase angles during two time periods of increased ITC: before fixations to the original location (Pre, $-420$ to $-380$ ms) and following fixations to the original location (Post, 80 to 120 ms) on Mismatch trials (for an example trial, see *Figure 3b*). As absolute phase angles can be difficult to interpret due to phase reversals caused by referencing (*Shirhatti et al., 2016*), we evaluated whether differences in phase between these time intervals were consistent for individual electrodes. We found significant differences ($p < 0.05$, uncorrected, permutation test) in 27 out of 32 electrodes, significantly more than expected by chance (binomial test, p <0.0001). Phase distributions for one example electrode are depicted in *Figure 3c*, showing peak-concentrated phase preceding fixations to the original object-location (mean = $-0.40$ rad, 95% CI [0.69–1.49], n = 179 fixations), and trough-concentrated phase following fixations to the updated location (mean = 2.85 rad, 95% CI [2.51 3.19], n = 244 fixations). Indeed, the average phase of these two distributions were significantly different ($Z_F = 10.3$, $p < 0.0001$, permutation test).

We next tested whether preferred phases were consistent across recording sites (*Figure 3d*). Average theta phase preceding fixations to the original object-location occurred near the peak of theta (mean = 0.2 rad, 95% CI [$-0.7$ 1.0]). In contrast, the average theta phase following fixations to the updated object-location occurred near the trough of theta (mean = 2.8 rad, 95% CI [2.3 3.3]). Permutation testing revealed consistent differences in phase across electrodes ($Z_F = 6.8$, permutation $p < 0.001$, n = 32 electrodes). While these phase differences between conditions appear to be driven by memory processing, it is possible that they resulted from differences in timing alone. Two pieces of evidence refute this conclusion. First, the distribution of phase in the interval preceding fixations to updated object-locations is indistinguishable from a uniform distribution ($Z = 1.04$, $p = 0.36$, Rayleigh test, n = 32 electrodes). Second, following fixations to the original object-location, we found phase concentrated around theta troughs (mean = 2.9 rad, 95% CI [2.5 3.4]). These phases were not distinguishable from those following fixations to the updated object-location ($F_{62} = 0.77$, $p = 0.38$, n = 32 electrodes). These results provide additional evidence that the timing of memory-dependent fixations depend upon the phase of theta.

## Theta phase consistency during updated-location viewing predicts subsequent memory

If the observed phase-locking of eye movements to hippocampal theta reflects encoding and retrieval operations, we would predict that the strength of coherence between visual sampling and

theta determines memory performance. To test this prediction, we tested whether ITC predicted subsequent memory performance (i.e. differed between trials where the original location was subsequently remembered or forgotten) on the final recognition test (*Figure 4*). We observed no significant subsequent memory effects following fixations to the original object-location (*Figure 4a*, nonparametric cluster test, $P_{FWE}$ = 0.38, n = 5 subjects). Conversely, phase-locking following fixations to updated locations during Mismatch trials predicted subsequent memory performance (*Figure 4b*). Greater ITC at frequencies ranging from 3 to 6 Hz following fixations to the updated location was associated with better memory for the original location on the final recognition test

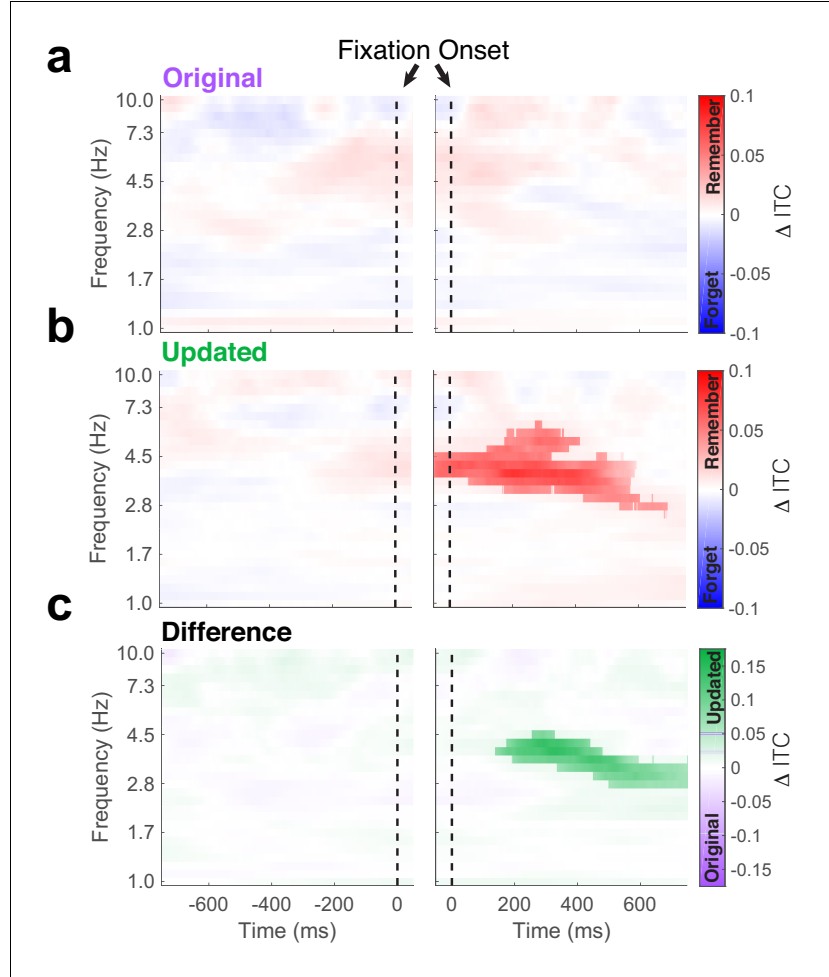

**Figure 4.** Phase-locking of hippocampal theta predicts subsequent memory. Time-frequency plots depict differences in inter-trial phase coherence (ITC) between subsequently remembered and forgotten Mismatch trials. (a) No subsequent memory effects were present during fixations to the original object-location. (b) Significant ($P_{FWE}$ < 0.05, nonparametric cluster corrected) increases in phase-locking were associated with memory following the fixations to the updated object-location. (c) Subsequent memory effects were specific to updated object-locations, as revealed by the significant ($P_{FWE}$ < 0.05, nonparametric cluster corrected) interaction following fixation onset.

The online version of this article includes the following source data and figure supplement(s) for figure 4:

**Source data 1.** MATLAB code and source files to reproduce data in *Figure 4*.

**Figure supplement 1.** Impact of saccades in pre- and post-fixation windows on inter-trial phase clustering.

**Figure supplement 1—source data 1.** MATLAB code and source files to reproduce data in *Figure 4—figure supplement 1*.

**Figure supplement 2.** Theta phase resets follow fixations to objects and predict subsequent memory.

**Figure supplement 2—source data 1.** MATLAB code and source files to reproduce data in *Figure 4—figure supplement 2*.

(nonparametric cluster test, $P_{FWE} < 0.0001$, $g = 0.5$, n = 5 subjects). The ability of hippocampal phase-locking to predict memory outcomes was specific to viewing updated object-locations; this subsequent memory effect was significantly greater following fixations to updated as opposed to original object-locations (*Figure 4c*, nonparametric cluster test, $P_{FWE} < 0.0001$, n = 5 subjects). Furthermore, these differences could not be accounted for by differences in power (all $P_{FWE} > 0.05$, nonparametric cluster correction, n = 5 subjects). These results suggest that the reset of hippocampal theta by novel sensory information may be a major determinant of memory performance on this task. Moreover, the pattern of viewing behavior on Mismatch trials indicates that the detection of the updated object-location precedes retrieval-guided viewing of the original object-location (*Figure 1d*), suggesting that retrieval-related changes in ITC were predictive of memory-guided visual exploration but not memory outcomes.

## Gaze-dependent theta modulations are consistent across subjects and electrodes

In the previously reported analyses of hippocampal phase and memory processes, we adopted the conservative approach of group-level inference (i.e. we tested for differences in the group mean while treating subject as a random factor), without focusing on variability in the observed effects across subjects or anatomical locations within the hippocampus. To supplement these findings, we report the electrode-level results of the main theta phase-locking analyses broken down by individual subjects (*Table 3*). While the magnitude of these effects is biased due to selection from significant group-level effects, we observed significant retrieval effects in about one third of all electrodes, and more than half showed significant novelty and subsequent memory effects. As such, the presented group results are common throughout our hippocampal recordings.

## Rapid sequential fixations do not account for memory-specific theta effects

Our ability to associate hippocampal theta with specific memory processes assumes a strong correspondence between theta phase dynamics and the moments surrounding a fixation of interest. However, eye movements made during task performance were necessarily sequential in nature, as subjects iteratively viewed multiple locations on the screen. Thus, it is possible that the observed effects were influenced by these iterative behaviors (e.g. repeated viewing of novel locations). Indeed, subjects made multiple fixations during the epochs in which we observed significant phase-locking. To determine whether the observed effects were truly related to the fixations of interest, rather than preceding or following eye movements, we repeated our phase-locking analyses and restricted analysis to fixation events with a clean pre- or post-fixation window of 100, 200, and 400 ms in duration. While longer fixation-free windows greatly reduced our power to detect effects (see *Supplementary file 1* for details on the number of eye movements contributing to each analysis), we replicated our major findings in these analyses. Notably, we observed significantly increased ITC preceding fixations to original vs. updated locations on Mismatch trials (*Figure 4—figure supplement 1a*). In addition, we observed similar increases in ITC following fixations to updated vs.

**Table 3.** Proportion of electrodes showing significant theta phase-locking effects

|  | N | Retrieval | Associative novelty | Subsequent memory |
|---|---|---|---|---|
| S1 | 8 | 0.13 | 0.5[*] | 0.63[*] |
| S2 | 4 | 0 | 0.75[*] | 0.5[*] |
| S3 | 6 | 0.5[*] | 0.5[*] | 0.33 |
| S4 | 8 | 0.5[*] | 0.75[*] | 0.75[*] |
| S5 | 6 | 0.5[*] | 0.33 | 0.33 |
| Group | 32 | 0.34[*] | 0.56[*] | 0.53[*] |

Proportion of electrodes showing significant phase-locking effects. p-Values were computed using a binomial test, based on the total number of electrodes (N) for a subject or group. [*]p < 0.05, Bonferroni corrected across three conditions.

repeated objects (*Figure 4—figure supplement 1b*), the magnitude of which predicted whether memory for the original object-location was maintained (*Figure 4—figure supplement 1c*). These results indicate that the observed phase effects are likely related to fixations of interest, rather than sequential behaviors.

## Reset of hippocampal oscillations during memory-guided eye movements is specific to theta

While our analysis of phase dynamics focused on theta frequencies, it is possible that broadband or higher frequency (e.g. beta and gamma activity) phase dynamics are related to memory updating and retrieval. We tested for the presence of phase resets caused by individual fixations. From average ERPs at each channel, we computed pre-fixation (−750 to 50 ms) and post-fixation (−50 ms to 750 ms) power. A phase reset at a given frequency band would be indicated by an increase in post- vs. pre-fixation power, as phase alignment in this period would lead to increased power in the average ERP. We found evidence for a reset of ongoing oscillations in the 8 to 13 Hz range following fixations to presented objects, irrespective of condition (*Figure 4—figure supplement 2*). We observed significant differences (cluster $P_{FWE}$ = 0.006, $t_4$ = −4.04, $g$ = −1.14) in the magnitude of this phase reset effect when comparing fixations to the original and updated object-location on Mismatch trials (*Figure 4—figure supplement 2a*). Moreover, averaged ERPs locked to fixations to the original object-location exhibited increased power in the pre- vs. post-fixation interval at frequencies below 8 Hz (*Figure 4—figure supplement 2a*, cluster $P_{FWE}$ = 0.004, $t_4$ = −3.7, $g$ = −1.6). The magnitude of this effect was marginally different from fixations to updated locations (cluster $P_{FWE}$ = 0.06, $t_4$ = −1.5, $g$ = −0.6). These findings provide additional support for coordination between theta phase and the deployment of retrieval-dependent eye movements, as phase consistency increased prior to these retrieval-guided fixations. Finally, phase resetting following fixations to the updated object-location on Mismatch trials (*Figure 4—figure supplement 2c*) marginally predicted memory for the original object-location (cluster $P_{FWE}$ = 0.09, $t_4$ = 2.1, $g$ = 1.08). We did not observe evidence for phase resets at frequencies above 13 Hz, near the upper border of the alpha band.

## Theta to gamma phase amplitude coupling predicts memory updating

Having identified a consistent relationship between theta phase and specific viewing behaviors, we examined the relationship between theta phase and the amplitude of high frequency (80–200 Hz) gamma band activity. Phase-amplitude coupling (PAC) between theta and gamma has been proposed as a mechanism for separating memory representations (*Hasselmo and Eichenbaum, 2005*), with supporting evidence in both animal models (*Tort et al., 2009*) and humans (*Axmacher et al., 2010*; *Heusser et al., 2016*; *Lega et al., 2016*; *Lisman and Jensen, 2013*; *Vaz et al., 2017*). We used the modulation index (MI) to quantify PAC between the phase of theta (ranging from 1 to 10 Hz) and gamma amplitude (*Tort et al., 2010*).

We focused our PAC analyses on three comparisons of interest: associative novelty, memory retrieval, and memory performance. Results from an example electrode are depicted in *Figure 5*, showing increases in PAC related to forgetting of the original object-location. For a given electrode (*Figure 5a*), we computed gamma amplitude as a function of theta phase during each trial type of interest. These measures were used to compute the MI, which measures PAC as the difference between the observed amplitude distribution and a uniform distribution (*Figure 5b*, dashed line). To make sure observed differences in PAC did not result from non-stationarities in the data or common task-evoked changes in amplitude and phase (*Aru et al., 2015*), we permuted phase information across trials for each condition and computed a normalized score ($MI_Z$) based on this null distribution. We assessed changes in $MI_Z$ across conditions (*Figure 5c*) to identify memory-related changes in PAC. Across subjects, we found 25% of electrodes exhibited significant differences in PAC driven by associative novelty (i.e. differences between fixations to updated and repeated locations), significantly more than expected by chance (binomial test, p < 0.001, proportion difference = 0.2). This effect was primarily driven by increased PAC during the viewing of updated locations. We found significantly greater PAC on 16% of electrodes ($P_{FWE}$ < 0.05, permutation test), compared to surrogate measures obtained by permuting theta phase across fixation events (significantly more than expected by chance, binomial test, p = 0.02). Only 9% of electrodes exhibited significant differences

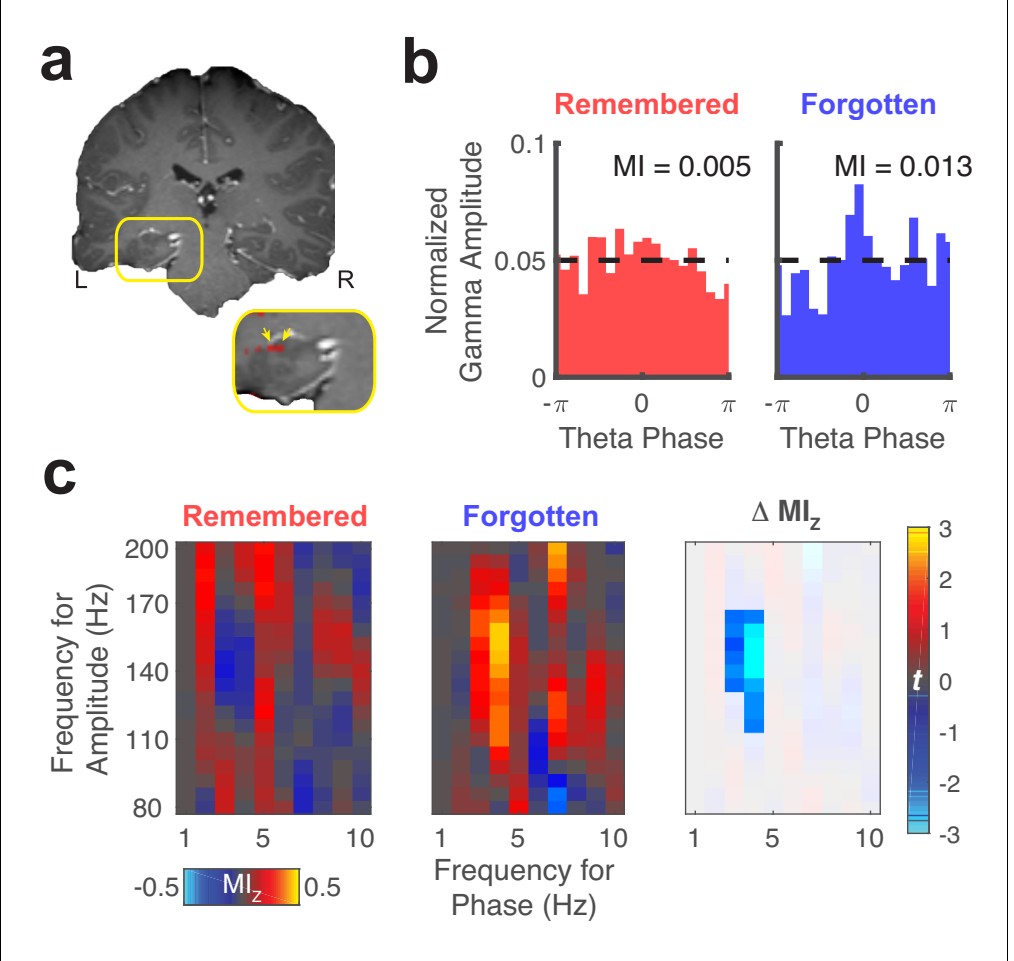

**Figure 5.** Representative theta to gamma phase amplitude coupling at an individual electrode. (**a**) Re-referenced bipolar recording from contacts in the left hippocampus and adjacent white matter. (**b**) Normalized amplitude distributions reveal memory-related modulation of gamma (150 Hz) amplitude by theta (4 Hz) phase at this recording site. MI, modulation index. Dashed line denotes normalized gamma amplitude under a uniform distribution. (**c**) Left, comodulograms depict increased PAC (z-scored MI, constructed from trial-shuffled surrogate data) during fixations to updated object-locations when memory for the original object-location was forgotten. Right, the statistical map depicts a cluster of significant ($P_{FWE} < 0.05$, nonparametric cluster corrected) cross-frequency interactions.

The online version of this article includes the following source data for figure 5:

**Source data 1.** MATLAB code and source files to reproduce data in *Figure 5*.

in PAC preceding fixations to the original versus updated locations on Mismatch trials (binomial test, p = 0.14, proportion difference = . 05).

Next, we tested for group-level differences in PAC during specific viewing behaviors. To account for variability in theta frequency across subjects and electrodes, we selected the theta frequency that exhibited the greatest magnitude $MI_Z$ from 4 to 6 Hz, irrespective of condition. We first examined if theta to gamma PAC was sensitive to associative novelty by contrasting PAC following fixations to updated versus repeated objects. We found significantly increased theta to gamma (80 to 100 Hz) PAC during fixations to updated object-locations (nonparametric cluster test, $P_{FWE}$ = 0.05, g = 1.2, n = 5 subjects), indicating that gamma amplitude was more dependent on theta phase when visual stimuli conflicted with memory (*Figure 6a*).

Given significant PAC effects during fixations to updated object-locations, we next evaluated whether PAC during these fixations predicted subsequent memory performance. We found that increases in theta to high-gamma (130 to 150 Hz) PAC were significantly greater (nonparametric

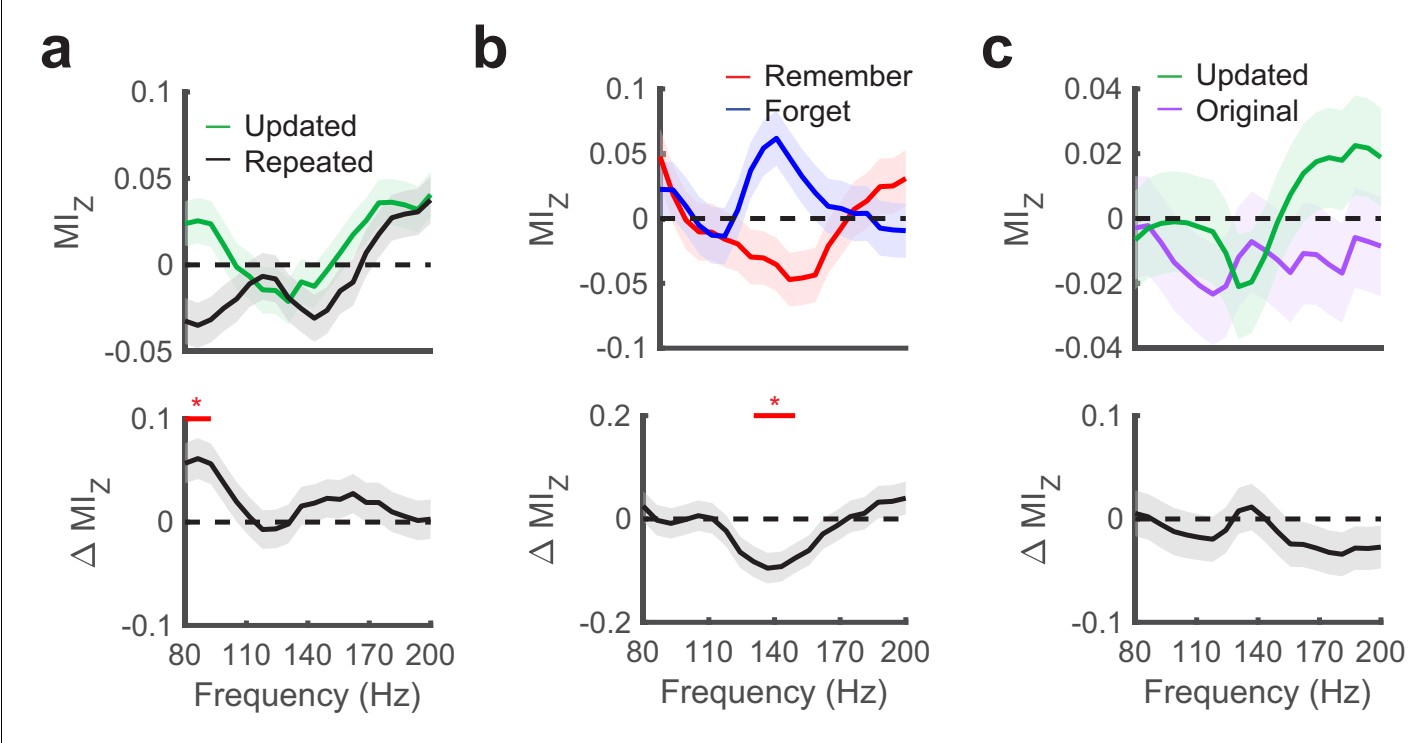

**Figure 6.** Hippocampal phase amplitude coupling predicts novelty detection and memory updating. (a) Top, post-saccade changes in PAC following (−50 ms to 750 ms) fixations to updated and repeated object-locations are displayed for a range of gamma amplitudes. Below, significant ($P_{FWE} < 0.05$, nonparametric cluster corrected) increases in PAC related to novelty are indicated. (b) Theta to gamma PAC during (−50 ms to 750 ms) fixations to updated object-locations varies with memory outcome. A significant ($P_{FWE} < 0.05$, nonparametric cluster corrected), negative subsequent memory effects is depicted in the bottom panel. (c) PAC did not differ in the moments leading up to (−750 ms to 50 ms) fixations to updated and original object-locations during Mismatch trials. List of Tables.

The online version of this article includes the following source data for figure 6:

**Source data 1.** MATLAB code and source files to reproduce data in *Figure 6*.

cluster test, $P_{FWE} = 0.04$, $g = −0.9$, n = 5 subjects) on trials where the original object-location was subsequently forgotten (*Figure 6b*). Because forgetting was caused by interference from updated object-locations (subjects chose the updated location on 73% (SD = 3%) of final recognition trials), these results indicate that increased theta to high-gamma PAC reflects updating of the object-location in memory. Finally, we tested whether PAC varied with retrieval demands by comparing measures of PAC in the moments preceding fixations to the original and updated object-locations on Mismatch trials (*Figure 6c*). We did not observe a consistent relationship between PAC and retrieval-related eye movements, consistent with our electrode-level analyses.

As we observed different gamma frequencies modulated by theta associated with novelty detection vs. subsequent memory, we aimed to compare the specificity of these effects directly. To do this, we compared the difference in $MI_z$ between the two frequency ranges for each contrast. This analysis suggested specificity of memory updating in the high-gamma range (130–150 Hz), as it produced greater subsequent memory effects compared to the 80 to 100 Hz gamma band associated with novelty detection (paired t-test, $t_4 = 2.6$, p = 0.06, g = 0.7). On the other hand, we did not observe this level of specificity associated with associative novelty (paired t-test, $t_4 = −0.2$, p = 0.84, g = −0.1). These results implicate increased theta to high-gamma PAC in the updating of previously formed memory traces.

We examined the specificity of the observed PAC effects to different theta frequencies by repeating the aforementioned analyses for low-theta (1–3 Hz) and faster theta/alpha frequencies (7–10 Hz). We did not observe significant memory-related differences in PAC in these ranges. Directly comparing PAC effects across different ranges of theta, we found significantly greater associative novelty-

related changes in PAC when using 4–6 Hz phase (identified via clear peaks in the power spectrum, see *Figure 2b*) to define the modulating frequency compared to the faster theta/alpha band (cluster $P_{FWE}$ = 0.01, $t_4$ = 1.2, $g$ = 0.7). There was weak evidence for stronger effects than the low-theta band (cluster $P_{FWE}$ = 0.08, $t_4$ = 1.9, $g$ = 1.3), suggesting specificity of associative novelty-related PAC in the 4–6 Hz theta range. No evidence for frequency specificity was found when comparing updating-related PAC effects across different ranges of theta (all cluster $P_{FWE}$ > 0.13). The observed interactions between theta phase and gamma amplitude likely reflect distinct processing states in hippocampal networks, wherein asynchronous local activity is necessary to segregate novel perceptual information from previously encoded memories.

## Memory-related changes in PAC are unrelated to theta waveform properties

Changes in theta to gamma PAC can arise from multiple sources such as the sharpness of non-sinusoidal oscillations, differences in oscillatory power, and phase-locking of ongoing oscillations to sensory events (*Cole et al., 2017*; *Cole and Voytek, 2017*; *Vaz et al., 2017*). We performed a series of control analyses to determine whether the observed statistical differences in PAC were likely to reflect a direct relationship between two distinct oscillatory features (i.e. modulation of gamma amplitude by theta phase) as opposed to other properties of theta oscillations. For each fixation of interest, we measured the average theta amplitude, sharpness of peaks and troughs, waveform asymmetry, and measures of phase-locking (the difference in phase angle between each trial and the average within-condition phase angle for each condition and time point). We then asked whether novelty- and updating-related changes in PAC ($MI_Z$) could be explained by changes in these properties of theta waveforms.

In our initial analysis of associative novelty (contrasting fixations to updated vs. repeated object-locations), we identified theta to gamma (80–100 Hz) PAC. We found that theta waveform properties were significantly related to PAC in this frequency range for a minimal number of electrodes (range 2 to 5 electrodes, see *Table 4* for details), indicating that there was no consistent relation between theta waveform properties and novelty-related PAC. Theta waveform properties did not influence theta to gamma (130–150 Hz) PAC associated with memory updating, with one notable exception (see *Table 5* for details). Changes in theta power predicted trial-level differences in PAC in 22% of electrodes, significantly more than expected by chance (p = 0.0008, binomial test).

We next asked whether changes in these theta waveform properties contributed to the observed changes in PAC associated with associative novelty and memory updating. To confirm that our PAC analyses were not primarily caused by differences in theta amplitude (or any other oscillatory features such as peak/trough sharpness), we repeated our initial PAC analysis after regressing out variability explained by additional single-trial measures of theta amplitude, phase-locking, and waveform shape. The results of this analysis revealed the same relation between PAC and detection of associative novelty and memory updating. These control analyses rule out the possibility that observed differences in PAC resulted from changes to theta waveform shape, which is non-sinusoidal

**Table 4.** Proportion of electrodes showing changes in PAC due to theta waveform properties while viewing all objects during refresh.

|  | N | Power | Sin($\theta$) | Cos($\theta$) | $S_{peak}$ | $S_{Trough}$ | Asym | Full |
|---|---|---|---|---|---|---|---|---|
| S1 | 8 | 0 | 0.13 | 0 | 0 | 0.13 | 0 | 0.13 |
| S2 | 4 | 0.50* | 0 | 0 | 0.25 | 0 | 0 | 0 |
| S3 | 6 | 0 | 0.33 | 0 | 0 | 0 | 0 | 0.17 |
| S4 | 8 | 0.25 | 0.13 | 0 | 0 | 0 | 0.25 | 0.13 |
| S5 | 6 | 0.17 | 0.17 | 0 | 0.17 | 0.17 | 0.17 | 0.17 |
| Group | 32 | 0.16 | 0.16 | 0 | 0.06 | 0.06 | 0.09 | 0.13 |

Proportion of electrodes showing significant modulation of PAC by properties of theta, including power, single-trial measures of phase-locking, peak and trough sharpness, and waveform asymmetry. Each column denotes the parameters included in each regression. Model significance was determined with the *F*-statistic. *p < 0.05, Bonferroni corrected across seven tests.

**Table 5.** Proportion of electrodes showing changes in PAC due to theta waveform properties while viewing Mismatch objects.

| | N | Power | Sin($\theta$) | Cos($\theta$) | $S_{Peak}$ | $S_{Trough}$ | Asym | Full |
|---|---|---|---|---|---|---|---|---|
| S1 | 8 | 0.25 | 0 | 0.13 | 0 | 0.13 | 0.38[*] | 0.13 |
| S2 | 4 | 0 | 0.25 | 0.25 | 0.25 | 0 | 0 | 0.25 |
| S3 | 6 | 0.33 | 0 | 0 | 0 | 0 | 0.17 | 0 |
| S4 | 8 | 0.13 | 0 | 0 | 0 | 0.25 | 0.13 | 0 |
| S5 | 6 | 0.33 | 0.17 | 0 | 0 | 0 | 0 | 0.17 |
| Group | 32 | 0.22[*] | 0.06 | 0.06 | 0.03 | 0.09 | 0.16 | 0.09 |

Proportion of electrodes showing significant modulation of PAC by properties of theta, including power, single-trial measures of phase-locking, peak and trough sharpness, and waveform asymmetry. Each column denotes the parameters included in each regression. Model significance was determined with the *F*-statistic. [*]p < 0.05, Bonferroni corrected across seven tests.

and known to vary with behavioral state (*Cole and Voytek, 2017*; *Scheffer-Teixeira and Tort, 2016*).

## Discussion

We set out to examine the relationship between hippocampal theta oscillations and memory-driven viewing behaviors during visual exploration. To achieve this goal, we examined simultaneously recorded hippocampal potentials and eye movements while neurosurgical patients performed a spatial memory task. This paradigm allowed the disambiguation of eye movements driven by associative novelty (i.e. objects presented in updated spatial locations) and memory retrieval. We discovered that these two distinct viewing behaviors were uniquely tied to theta: phase-locking preceded retrieval-guided eye movements and followed novelty-driven fixations, predicting memory performance. These retrieval effects primarily occurred at the peak of theta, whereas novelty effects occurred at the trough of theta. Thus, we provide empirical support for models suggesting that interference between sensory information and memory representations is avoided by timing encoding and retrieval to occur at distinct phases of theta (*Hasselmo et al., 2002*). Analysis of theta to gamma PAC confirmed a potential mechanism by which the theta rhythm supports both encoding and retrieval. Modulation of gamma (80–100 Hz) amplitude by the theta rhythm increased during detection of updated object-locations. Further, increased theta to high-gamma PAC predicted memory updating, as determined by a subsequent memory test. These data support theories of active hippocampal involvement in visual exploration (*Voss et al., 2017*) and provide novel evidence that theta oscillations support both retrieval- and novelty-dependent viewing behaviors.

Based upon electrophysiological studies in humans and primates (*Hoffman et al., 2013*; *Jutras et al., 2013*), it has become apparent that hippocampal memory representations are used to guide saccades to behaviorally relevant locations (*Meister and Buffalo, 2016*). Of particular relevance to the present work, visual exploration of novel, but not repeated, scenes leads to a reset of hippocampal theta oscillations (*Jutras et al., 2013*). The consistency of this hippocampal phase reset predicts the success of novel encoding. In the present work, we provide evidence for conserved hippocampal processing in humans, by demonstrating increased phase consistency of theta oscillations following fixations to novel locations, which were further associated with improved memory performance. Recent work using fMRI (*Liu et al., 2017*) also suggests that visual exploration of scenes is associated with hippocampal function, as the number of fixations during encoding predicted both hippocampal activity and subsequent memory. This relationship between hippocampal activity and viewing behavior was limited to novel (but not repeated) stimuli, suggesting hippocampal activity and exploratory eye movements are linked specifically when encoding novel content. Using our spatial memory paradigm to identify retrieval-dependent eye movements, we found theta phase-locking occurred prior to saccade onset. Taken together with previous studies that have tied hippocampal activity to retrieval-guided eye movements (*Bridge et al., 2017*; *Hannula and Ranganath, 2009*)

these results provide evidence that visual exploration is dependent upon the interplay of separate retrieval and novelty-detection mechanisms that underpin learning.

With the high temporal resolution of eye tracking and hippocampal potentials, our data are uniquely suited to clarify the hippocampal mechanisms that drive learning under associative novelty. Notably, we found increased theta phase-locking and power following fixations to mismatched spatial locations, consistent with previous hippocampal recordings in humans (*Chen et al., 2013*). The consistent timing of these effects following fixation onset provide further evidence that hippocampal theta is involved in the computation of mismatch signals, as opposed to providing signals of perceptual familiarity that could guide viewing to novel locations. This extends previous fMRI work which has established the involvement of the hippocampus in associative novelty detection (*Kumaran and Maguire, 2006*; *Kumaran and Maguire, 2007a*) and binding (*Bridge and Voss, 2014b*), specifically supporting the idea that the hippocampus acts as a comparator between new sensory inputs and prior memory (*Kumaran and Maguire, 2007b*; *Lisman, 1999*; *Lisman and Grace, 2005*; *Vinogradova, 2001*).

The observed interaction between the phase of hippocampal theta and amplitude of gamma oscillations provides insight into the mechanisms by which learning occurs under associative novelty, wherein sensory information conflicts with memory. Many theories propose that theta cycles segregate neuronal activity into functional packets based on coactivation of neuronal populations during distinct time windows (*Buzsáki and Moser, 2013*; *Hasselmo et al., 2002*; *Rennó-Costa and Tort, 2017*). These models allow for interleaved encoding and retrieval operations to occur at distinct phases of theta (*Colgin, 2015*), supported by nested higher frequency activity. In support of these models, we found that phase-locking associated with retrieval and novelty initiated at the peak and trough of hippocampal theta, respectively. In addition, theta to high-gamma PAC increased during the detection of novel spatial information, consistent with theta oscillations organizing high-frequency computations across hippocampal networks. In doing so, we show these models explain when memory-driven eye movements occur in humans, particularly when there are demands for memory encoding and retrieval to co-occur.

We found theta to gamma PAC predicted whether previously encoded memories would be updated to reflect current sensory information, in opposition to task demands. Prior work has shown that increased theta to gamma PAC in hippocampus reflects successful encoding (*Lega et al., 2016*). These findings may explain the cause of memory updating in our task, which was associated with forgetting of the original object-location. If hippocampal PAC supports encoding, increased PAC could strengthen associations between objects and updated locations. During the final memory test, the strength of these associations interferes with memory for the original location, resulting in forgetting. Despite this account, we cannot rule out the possibility that multiple processes contribute to memory updating. For example, updating could result from weakening of older memories during new learning or biased competition between memories at retrieval (*Kuhl et al., 2012*). Although our experimental design cannot distinguish between these accounts (i.e. memory updating necessarily causes forgetting of the original object-location), our findings clearly demonstrate that increased PAC predicts the formation of novel memories at the expense of prior learning.

Evaluation of theta waveform properties revealed that memory-related differences in PAC resulted from nested oscillations, as opposed to amplitude modulation of sharp waveforms or other changes to the theta waveform. Distinct neurophysiological states, defined by theta phase and modulation of gamma amplitude, indicated when viewing was driven by memory as opposed to novel information in the environment. While we did not observe theta-dependent modulation of gamma amplitude during retrieval, this could result from a number of methodological constrains, such as biased sampling to the anterior hippocampus and potential obscuring of narrowband gamma signals commonly observed in microelectrodes due to summation across larger neuronal populations.

Despite the emerging consensus in the rodent that theta oscillations in the hippocampus are responsible for segregating neuronal computations involved in encoding and retrieval operations (*Colgin, 2015*; *Colgin, 2016*; *Hasselmo et al., 2002*), translating these findings to observations in humans has proven challenging. Comparative studies between rodents and humans suggest that theta rhythms in the human are commonly observed at lower frequencies (e.g. 3 vs. 8 Hz) during exploratory behaviors (*Watrous et al., 2013a*). Additionally, studies have demonstrated multiple sources of theta rhythms in the medial temporal lobe (*Lega et al., 2012*; *Mormann et al., 2008*), including a low-theta or delta band (1–4 Hz) in addition to the typical theta band (4–8 Hz). Based on

observations that encoding-related increases in power (*Burke et al., 2013*; *Lega et al., 2012*; *Miller et al., 2018*) and PAC (*Lega et al., 2016*) are specific to the low-theta band, it has been proposed that lower frequency theta in humans reflects a homologue of rodent theta (*Jacobs, 2014*). In contrast to this body of work, we observed faster (i.e. predominantly greater than 4 Hz) memory-related theta dynamics. We believe this difference stems from the emphasis on visual information in our task, resulting in a task-dependence in the frequency of theta oscillations. Higher frequency theta effects have been observed during visual search (*Hoffman et al., 2013*), in which alignment between saccades and theta oscillations was focused between 6–8 Hz. Our findings that fixations to objects caused theta phase resets in the 6–12 Hz range are consistent with work in nonhuman primates (*Jutras et al., 2013*), which demonstrated that saccades during visual exploration caused resets in hippocampal oscillations predominantly in the 8–11 Hz range. As in the present study, the magnitude of these resets predicted the success of encoding during memory formation. Multiple factors could account for changes in the speed of hippocampal theta, including the type of representation being processed in the medial temporal lobe (*Watrous et al., 2013b*), spatial attention required by a given task, or the rate of fixations. Future studies are necessary to determine what factors determine the speed of hippocampal theta oscillations and their relevance to different forms of memory.

Our findings of theta-dependent eye movements in the hippocampus are relevant to models of functional organization across the hippocampus. Drawn primarily from animal models, theories of hippocampal organization emphasize functional segregation along its long axis (*Fanselow and Dong, 2010*; *Strange et al., 2014*), with general agreement regarding a role of the anterior (ventral in rodents) hippocampus in emotion and affect, and the posterior (dorsal in rodents) hippocampus in spatial navigation and memory. As in most iEEG studies of hippocampal function in humans, we recorded primarily from the head and body of the hippocampus (see *Figure 1b*). Given the demands for spatial and mnemonic processing during our task, our results are not easily accommodated by these models. Our findings are consistent with recent theories that emphasize gradients in the scale of representations along the hippocampal long axis (*Bellmund et al., 2018*; *Poppenk et al., 2013*) with transitions from general to precise representations in the anterior to posterior direction. Thus, the emphasis of global spatial relations in our task (i.e. the relative positions of the original and updated locations rather than precise locations) may account for the observed effects within anterior hippocampus. Because our coverage was limited to the anterior aspect of the hippocampus, our study cannot directly address the anatomical specificity of memory-guided eye movements. However, modulation of hippocampal theta by eye movements has also been observed in the anterior hippocampus of non-human primates (*Jutras et al., 2013*), suggesting a conserved mechanism across primates.

While our analysis of theta oscillations was restricted to electrodes in the hippocampus (or adjacent white matter), memory-guided exploratory behaviors depend upon interactions between distributed cortical systems (*Voss et al., 2017*), particularly those involved in representing features within a scene, the spatial relations between these features, and transforming these memory representations into an oculomotor plan based on current visual input. Cells within the entorhinal cortex of macaques code the location of fixations in a grid-like fashion during free viewing (*Killian et al., 2012*), serving as a potential mechanism to provide a scale-invariant representation of fixation locations within the scene (*Bicanski and Burgess, 2019*). Similar grid-like modulation of entorhinal activity has been observed in humans using fMRI (*Julian et al., 2018*; *Nau et al., 2018*), providing converging evidence across species that the entorhinal system may provide a spatial framework for memory-guided viewing. As such, synchronous theta oscillations between the hippocampus and entorhinal systems would provide the spatial coding necessary to inform the oculomotor system of memory-relevant information. Although we were limited by electrode coverage, examining entorhinal-hippocampal synchrony and interactions between the hippocampus and other cortical systems should be a key aim for future studies.

One potential caveat is that the observed eye movements in this study do not reflect natural exploratory behaviors per-se but are rather driven by demands to learn and maintain the original object-location throughout the task. Thus, it is unclear how stereotyped these retrieval-dependent eye movements would be during unconstrained visual exploration. Free-viewing paradigms could build on this theoretical framework and determine the extent to which hippocampal-dependent viewing behaviors occur without task constraints. Given the observed retrieval phase-locking effects

occurred well before saccade initiation, it is likely that the hippocampus plays a causal role in generating these eye movements. Causal manipulation of hippocampal theta, including stimulation-based approaches could be used to test this hypothesis, which is supported by growing evidence for disruptions in viewing behaviors from amnesic patients with hippocampal damage (*Hannula et al., 2007*; *Lucas et al., 2019*; *Olsen et al., 2016*; *Ryan et al., 2000*; *Smith et al., 2006*).

In conclusion, encoding and retrieval dependent eye movements are time locked to the phase of the hippocampal theta rhythm. Our findings support models wherein distinct phases of the theta cycle segregate neural processing of information related to encoding and retrieval (*Colgin, 2016*; *Hasselmo et al., 2002*; *Hasselmo and Eichenbaum, 2005*). Akin to spatial attention shifting between multiple locations relevant to a task at hand (*Landau et al., 2015*; *Re et al., 2019*), hippocampal theta could coordinate visual sampling between novel content in the environment and memory-rich spatial locations. The hippocampus thereby contributes to memory-guided behaviors through coordinated sampling of current and past perceptual states.

## Materials and methods

### Participants

Five subjects (three male; see *Table 6* for demographic information) with refractory epilepsy performed our associative memory task during their stay at Northwestern Memorial Hospital (Chicago, IL). All subjects had depth electrodes implanted in the hippocampus as part of neurosurgical monitoring prior to elective surgery. Written informed consent was acquired from all subjects prior to participation in the research protocol in accordance with the Northwestern University Institutional Review Board.

### Experimental paradigm

We tested memory for associations between objects and their spatial locations using a novel spatial memory task. This task consisted of three distinct phases (Study, Refresh, Recognition), with each phase separated by a 60 s distractor (free viewing of scenes with domestic felines; *Zhang et al., 2008*). Subjects performed eight blocks in which they learned spatial locations for a sequence of 16 unique objects. Eye movements were recorded during each phase of the task, with five-point gaze calibration performed before each phase. Objects were 128 trial-unique images of real-life objects from the Bank of Standardized Stimuli (*Brodeur et al., 2010*). During each phase of the task, objects were presented at 3° of visual angle, with a red square of 0.2° of visual angle centered on each object. Stimuli were presented on a 23.6" monitor with a 120 Hz refresh rate from a stimulus control laptop. Synchronization pulses were sent from the stimulus control laptop to the clinical recording system using a DAQ control board, allowing alignment of electrophysiological and behavioral data.

At the beginning of each Study phase, a unique background image appeared for 5 s. Scenes provided visual information to assist learning unique spatial locations of 16 objects presented in the following block. Throughout the remainder of the Study phase, a sequence of 16 objects were

**Table 6.** Subject demographics.

|  | S1 | S2 | S3 | S4 | S5 |
|---|---|---|---|---|---|
| Age (years) | 20 | 34 | 53 | 25 | 44 |
| Sex | M | M | M | F | F |
| Full-scale IQ | 94 | 109 | 105 | 121 | 92 |
| Implanted Hemisphere | Left | Left | Right | Left | Left |
| Epileptic Focus | Basal temporal | Basal temporal | Middle hippocampus | Basal temporal | Amygdala |
| Etiology | Cortical dysplasia | Dysembryoplastic neuroepithelial tumor | Focal cortical dysplasia | Low grade glioma | Mesial temporal sclerosis |
| Duration of epilepsy (years) | 10 | 10 | 8 | 3 | 41 |
| Hippocampal contacts (n) | 8 | 4 | 6 | 7 | 6 |

presented at distinct locations superimposed on the background scene. At the start of each study trial, a fixation cross flashed twice on the screen (250 ms per flash, separated by 250 ms of the background scene) at the location of the next object. The fixation cross remained on the screen for a duration of 2 s, followed by presentation of the object for 3 s.

Next, subjects were tested on their spatial memory for each of the objects during the Refresh phase. During this phase of the task, three location cues indicated by small red squares (0.2° of visual angle) were presented in an equilateral triangle (randomly selected distance for each stimulus, mean distance of 12° and a range of 5.9–21.1° of visual angle across presented arrays). The object was presented at one of these three locations. Importantly, one of these locations was the object's original location. On each block, half of the trials were randomly assigned to the Mismatch condition, in which the object was presented at one of the two novel locations. On the Match trials, the object was presented in its original location. Each trial began with the presentation of the background scene for 1 s followed by a fixation cross at the center of the screen for 1 s, at which point the object and location cues appeared for 5 s. Following stimulus presentation, memory for the original location of each item was tested. Subjects determined whether the object was in its original location, a new location, or if they were unsure by clicking a box that said: Same, Different, or Unsure. No feedback was given regarding the accuracy of each response.

Each block concluded with the Recognition phase which served as a final memory test for the original object-locations. The background scene appeared for 1 s, followed by the presentation of a fixation cross in the center of the screen for 1 s. Then, each object was presented at all three locations for a duration of 5 s. Following stimulus presentation, subjects selected the original object-location using a three-button response pad.

## Eye tracking

Eye movements were recorded at 500 Hz using an Eyelink 1000 remote tracking system (SR Research, Ontario, Canada). Continuous eye-movement records were parsed into fixation, saccade, and blink events. Motion (0.15°), velocity (30°/s) and acceleration (8000°/s$^2$) thresholds were used to identify saccade events. Blinks were identified based on pupil size, and remaining epochs below detection thresholds were classified as fixations. The location of each fixation event was computed as the average gaze position throughout the duration of the fixation. Circular viewing regions of interest (ROIs) were constructed based on a distance of 6° from one of the three potential object-locations. We focused our analysis of hippocampal activity to the subset of fixation events greater than 80 ms in duration. Except for our analyses of associative novelty (fixations to updated object-locations vs. repeated object-locations), which focused on early fixations to objects presented either in the original or updated object-location, we restricted our analysis to fixations that occurred 500 ms after object presentation and 500 ms before the end of each trial to avoid stimulus onset and offset effects.

## Intracranial recordings

A combination of depth electrodes (Integra Life Sciences, Plainsboro NJ; AD-TECH Medical Instrument Co., Racine, WI; DIXI Medical, Besançon, France) as well as subdural grids and strips were implanted for neurosurgical monitoring. Our analyses focused on hippocampal depths, which had electrodes spaced 5 mm apart. Electrophysiological data were recorded to a clinical reference using a Nihon Kohden amplifier with a sample rate of 1–2 kHz with a bandpass filter from 0.6 to 600 Hz. Data were re-referenced to a bipolar montage and downsampled to 500 Hz as part of preprocessing. We analyzed bipolar pairs with at least one contact in hippocampal grey matter or proximal white matter. Line noise was reduced by application of a band-stop 4th order Butterworth filter. To rule out the possibility that epileptiform activity influenced our analyses, electrodes that exhibited inter-ictal spiking were excluded from analysis. In addition, all analyses were repeated after excluding contacts within the seizure onset zone (two electrodes in S3). The observed results were qualitatively identical, with no statistical differences when including all electrodes (all p > 0.05).

## Anatomical localization

Post-implant CT (n = 4) or T1 weighted structural images (n = 1) were coregistered with presurgical T1 weighted structural MRIs using SPM12. Subdural electrodes were localized by reconstructing

whole-brain cortical surfaces from pre-implant T1-weighted MRIs using the computational anatomy toolbox (*Dahnke et al., 2013*) and snapping electrode centroids to the cortical surface based on energy minimization (*Dykstra et al., 2012*). All T1-weighted MRI scans were normalized to MNI space by using a combination of affine and nonlinear registration steps, bias correction, and segmentation into grey matter, white matter, and cerebrospinal fluid components. Deformations from the normalization procedure were applied to individual electrode locations identified on post-implant CT images or structural images using Bioimage Suite (https://medicine.yale.edu/bioimaging/suite/).

## Spectral decomposition

To examine oscillatory processes in the hippocampus, we decomposed bipolar recordings into measures of spectral phase and power using the continuous Morlet wavelet transform (wave number 5) across 30 logarithmically spaced frequencies from 1 to 10 Hz. We examined 1500 ms windows surrounding each fixation event of interest, with a 1250 ms buffer to prevent edge artifacts. Additional analysis of power spectra used the multitaper method to estimate spectral densities, ranging from 1 to 250 Hz. To identify oscillations, we modeled the power spectra as a mixture of two components: an aperiodic component modeled with an exponential, and putative oscillatory components modeled with Gaussians (*Haller et al., 2018*). The presence of oscillatory components was then evaluated by testing the equality of variances between aperiodic and full models.

## Phase-locking analyses

We examined the relationship between the hippocampal theta rhythm and individual eye movements by computing the inter-trial phase coherence (a measure of phase-locking):

$$ITC_{ft} = \frac{1}{N} \left| \sum_{k=1}^{N} e^{i\varphi ftk} \right|$$

for a given time (*t*) and frequency (*f*), where *N* is the total number of individual trials, *k*, and $e^{i\varphi}$ is the polar representation of the phase angle, $\varphi$. This measure was computed separately for individual conditions of interest (e.g. fixations to a specific region of interest on the display). As this measure is biased by the number of observations, with fewer observations leading to inflated ITC measures, we used a random subsampling approach to ensure that the number of observations were matched prior to statistical testing. In addition to examining differences in the magnitude of phase-locking, we compared differences preferred phase angles, across conditions of interest. We estimated a difference in phase angles using the Watson-Williams test (*Berens, 2009*), followed by nonparametric statistical testing to assess significance.

## Phase amplitude coupling analyses

Cross-frequency coupling between the phase of theta and gamma amplitude was computed using MI (*Tort et al., 2010*). MI is defined as the deviation in an amplitude distribution (across phases) from a uniform distribution, an adaptation from Kullback-Leibler distance (*Kullback and Leibler, 1951*), $D_{KL}$, that normalizes the range of the distance between zero and one:

$$MI = \frac{D_{KL}(P, U)}{log(N)}$$

where *P* is the normalized amplitude distribution as a function of phase, *U* is a uniform distribution, and *N* is the number of phase bins. For all presented analyses, we used 20 phase bins of 18°. MI takes values greater than zero when the observed amplitude varies with phase and is equal to zero when the distribution is uniform.

To circumvent the relatively short epochs in which we analyzed cross-frequency coupling (constrained by the frequency of eye movements during our task), we computed a standardized measure of the modulation index, $MI_Z$, via a surrogate control analysis. Specifically, for each trial and frequency combination, we permuted the observed phase timeseries across trials (separately for each condition of interest). This procedure was repeated 1000 times, resulting in a null distribution of MI values that could be explained by random (or condition-evoked) variations in the observed signal

rather than true coupling between theta phase and gamma amplitude. $MI_Z$ was measured as the difference between the observed MI and mean of the surrogate distribution, in units of standard deviations. These measures were used for all subsequent analysis of cross-frequency coupling.

We performed a series of follow-up analyses to rule out the possibility that memory-related changes in PAC were driven by changes in oscillatory power, phase-locking, or changes in the shape of theta oscillations. For each fixation of interest, we filtered (4th order Butterworth filter) bipolar signal into theta (1 to 10 Hz). Peak and trough amplitudes were extracted from the theta signal. Peak and trough sharpness were extracted as the average change in amplitude 2 ms before and after the inflection point. Theta amplitude and phase were extracted from the Hilbert transform of the filtered signal. As a single-trial estimate of phase-locking, we computed the difference in theta phase for each event from the average phase for each condition of interest. Trial-level measures were computed by averaging across all peaks or troughs within a given time window.

We used multiple regression to identify potential linear relations between properties of theta waveforms and trial-level measures of PAC (i.e. $MI_Z$). Phase angles were transformed into linear components via sine and cosine transform prior to regression. In addition to assessing the significance of these relations, we computed an adjusted measure of $MI_Z$ that was independent of these theta waveform properties (the residuals of the linear model). Statistical evaluation of PAC was repeated after removing variance related to theta waveform properties.

## Statistical analyses

We adopted a nonparametric permutation-based approach (*Maris and Oostenveld, 2007*) to correct for multiple comparisons across time and frequencies. When comparing differences in ITC or power between different types of fixations, we constructed a null distribution of differences by permuting the assignment of condition labels, blocked at the subject level. This null distribution was used to define an independent cluster-forming threshold for each observed measure (e.g. ITC at a specific time-frequency pair). When measures would be biased by the number of observations per condition (e.g. differences in ITC), random subsampling was used to equate the number of observations per condition. Individual clusters were considered significant ($P_{FWE} < 0.05$) if the summed statistic within each observed cluster exceeded 95% or 97.5% of clusters in the null distribution for one- and two-tailed tests, respectively. For tests comparing the relationship between the phase of an oscillation and spectral power, null distributions were constructed by permuting the phase timeseries across trials within each condition, per electrode and subject. These null distributions were used to standardize measures of phase amplitude coupling prior to statistical testing, as described above.

This permutation procedure also determined statistical significance of oscillations present at individual electrodes. After computing an *F*-statistic measuring the variance in aperiodic and full power spectra (see *Spectral decomposition*), we generated permutation distributions (n = 10,000) by randomizing condition labels (i.e. raw or aperiodic signals) across fixations. Statistical tests at the subject and group level were performed using binomial tests, comparing the proportion of significant electrodes to chance (determined to be 0.05 by the null distribution). We set a threshold of p < 0.05 for statistical significance, using Bonferroni correction to control for multiple comparisons across conditions and time periods.

Our sample size was determined based on prior work in humans and nonhuman primates (*Hoffman et al., 2013*; *Jutras et al., 2013*; *Staudigl et al., 2017*), which reported robust modulations in hippocampal theta due to visual sampling with a sample sizes ranging from two to six subjects. To support future meta-analytic work, we report Hedges' *g* for dependent samples (*Hentschke and Stüttgen, 2011*) and proportion differences as measures of effect size where relevant. For cluster-based statistics, we estimated effect size by computing average effects within significant clusters on a per-subject basis (i.e. averaging across electrodes, frequency, and/or time) before group-level analysis.

## Acknowledgements

We are grateful to Dr. Christina Zelano for helpful discussions, Irena Bellinski for assistance with patient recruitment, and the Laboratory of Human Neuroscience at Northwestern University for sharing resources; DJB acknowledges the support of NIH/National Institute of Mental Health (NIMH; grant R21MH115366) and National Center for Advancing Translational Sciences, Grant Number

UL1TR001422. JEK was supported in part by National Institute of Neurological Disorders and Stroke grant T32NS047987.

# Additional information

## Funding

| Funder | Grant reference number | Author |
|---|---|---|
| National Institute of Mental Health | R21MH115366 | Donna J Bridge |
| National Center for Advancing Translational Sciences | UL1TR001422 | Donna J Bridge |
| National Institute of Neurological Disorders and Stroke | T32NS047987 | James E Kragel |

The funders had no role in study design, data collection and interpretation, or the decision to submit the work for publication.

## Author contributions

James E Kragel, Software, Formal analysis, Investigation, Visualization; Stephen VanHaerents, Jessica W Templer, Stephan Schuele, Joshua M Rosenow, Resources; Aneesha S Nilakantan, Investigation; Donna J Bridge, Conceptualization, Formal analysis, Supervision, Funding acquisition, Investigation, Methodology

## Author ORCIDs

James E Kragel https://orcid.org/0000-0002-3240-6203
Donna J Bridge https://orcid.org/0000-0001-9838-3094

## Ethics

Human subjects: Both verbal and written informed consent was obtained from all subjects prior to participation. This work was done in accordance with Northwestern University Institutional Review Board (IRB #: STU00202828).

## Decision letter and Author response

Decision letter https://doi.org/10.7554/eLife.52108.sa1
Author response https://doi.org/10.7554/eLife.52108.sa2

# Additional files

## Supplementary files

- Supplementary file 1. Number of fixations for each comparison of interest.
- Transparent reporting form

## Data availability

Behavioral data, eye movement data, continuous EEG recordings, and electrode locations in MNI space have been uploaded to NDAR.

The following dataset was generated:

| Author(s) | Year | Dataset title | Dataset URL | Database and Identifier |
|---|---|---|---|---|
| Kragel JE, VanHaerents S, Templer JW, Schuele S, Rosenow JM, Nilakantan AS, Bridge | 2019 | Simultaneous eye tracking and hippocampal iEEG to identify oscillatory signals of memory retrieval and novelty detection in humans | https://nda.nih.gov/edit_collection.html?id=2890 | NIMH Data Archive, 2890 |

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
