## [Decision Letter]

**Acceptance summary:**

This paper reports that eye movements are phase-locked to hippocampal theta rhythms in humans performing a spatial memory task. A major strength of the paper is its close link to theoretical predictions of distinct hippocampal theta phases being linked to novelty-related exploration/encoding and retrieval that have previously been supported by rodent data. The results provide quantitative data supporting this model in human subjects.

**Decision letter after peer review:**

[Editors’ note: the authors submitted for reconsideration following the decision after peer review. What follows is the decision letter after the first round of review.]

Thank you for submitting your work entitled "Hippocampal theta coordinates memory processing during visual exploration" for consideration by *eLife*. Your article has been reviewed by a Senior Editor, a Reviewing Editor, and three reviewers. The reviewers have opted to remain anonymous.

Our decision has been reached after consultation between the reviewers. Based on these discussions and the individual reviews below, we regret to inform you that your work will not be considered further for publication in *eLife*.

Reviewers all found the topic and experimental design compelling. However, all had significant concerns about the statistics and analyses that raised questions about whether the paper's conclusions were strongly supported by the results. The separate reviews are included below in their entirety, but major concerns that led to the rejection decision include:

1) The time-window for analysis: It is unclear that the data only include the fixation of interest (and not any other saccade or fixation). Changing this window would dramatically decrease the available data for analysis to ~250 ms for each fixation.

2) Improper statistics: Analyses lacked a direct test of the interaction of interest (and instead concluded differential effects based on the finding that one result was significant, and another was not).

3) The PAC results were viewed as a weak part of the paper. Concerns were raised that PAC effects may have resulted from changes in power or waveform asymmetry and that analyses were not corrected for multiple comparisons.

Differences in the gamma frequency observed (very fast) compared to previous results in rodents (e.g., Colgin et al.) made the discussion of the results as related to slow and fast gamma strange.

4) There was a lack of demonstration of peaks in the spectra for the frequency ranges that were chosen for analysis to justify why these particular frequencies were selected. It is possible that similar results observed at other frequencies and that drawing conclusions about specific oscillatory patterns are unwarranted.

5) Nothing is shown to describe inter-subject variability.

6) Pre- and post- fixation results were not shown for Match trials (i.e., similar to Figure 2).

If you decide to re-do your analyses and statistics, and the conclusions are supported by the new analyses, we would be willing to reconsider a resubmission of this manuscript.

*Reviewer #1:*Kragel et al. examine the relation between hippocampal field potentials and eye movements during a visuospatial memory task. The primary claims are that hippocampal theta oscillations at 5Hz exhibit phase-locking around the time of eye movements. Specifically, pre-saccade phase-locking increases for saccades to remembered locations whereas post-saccade phase-locking increases for new (updated) locations. They also reported that saccades to the locations of new stimuli were associated with increases in phase-amplitude coupling between theta and gamma activity.

I think the general topic of this paper is interesting but I have quite substantial concerns about whether the reported data back up the claims. Not all the findings are reported in sufficient detail and I see a number of statistical issues.

- The most substantial concern I see with the paper is that many of the paper's claims hinge on a comparison of two statistical tests, with the authors making much of the fact that one comparison is significant and that another is not. Instead the paper needs to directly test whether one effect is larger than the other, such as by testing the interaction with an ANOVA or some other way. This is a problem in many sections of the paper and substantially undermines the specificity of key claims. The authors should review all their claims and ensure that result is supported by a specific statistical test/interaction, rather than just relying on an effect being significant in one condition and not in another statistically.

- I was disappointed that the paper only measured the magnitude of phase-locking, while seemingly ignoring the key issue of what specific phase the locking occurred to. The models that they refer to from Hasselmo and Colgin have strong hypotheses related to specific theta phases and the recorded data would seem to measure this data. But unfortunately, the reported data analyses ignore testing which theta phase the locking occurs to, instead focusing only the magnitude of locking. Without this information, the paper doesn't really provide strong support for those theoretical models.

- It is notable that the latencies of the effects are close to 300ms post saccade. Are the effects related to P300s?

- The PAC analyses are not compelling because they do not rule out the possibility that the effects are driven by changes in power. Based on the data that they show, I think it remains possible that their PAC results are a direct result of changes in theta power or locking without any specific PAC changes. To substantiate their claim that their findings demonstrate a specific role for PAC in memory, the authors need to do much more to specifically show that their effects are caused by interactions between the timing of gamma and theta rather than power or phase changes in just one of the signals. They also should rule out whether the PAC changes could be related to waveform asymmetry.

- I had a hard time following the control analyses in the third paragraph of subsection “Theta dependence of memory-guided eye movements”. Can the text here be reworded so the logic is clearer?

- I was surprised to see high-gamma effects at frequencies as high as 130-150 Hz. Were these results corrected for multiple comparisons? In the Discussion section this pattern is compared to the fast and slow gamma oscillations seen in rodents by Colgin et al. However, this signal is so substantially faster in frequency than the signals seen in rodents that this comparison seems far-fetched.

- It seems that most of the paper's statistics are performed across subjects (based on the degrees of freedom). This is a useful, conservative approach, which they should better explain this in the results. They used a different approach in subsection “Theta to gamma phase amplitude coupling predicts memory updating”, when they report percentages of hippocampal electrodes that show each effect, which they should explain and justify.

Reviewer #2:

This manuscript uses an innovative analysis of memory-guided eye movements to examine the relationship between theta phase and encoding and retrieval processes through recordings from macro electrodes in the hippocampus in epilepsy patients. The authors report theta phase-locking prior to fixations to retrieved locations along with theta phase-locking following fixations to novel locations. Further, this phase-locking for novel (conflicting) information predicted memory for the original information. The authors also examined theta-gamma phase amplitude coupling (PAC) and identified increased PAC for fixations to updated locations compared to repeated locations, and that greater PAC was associated with worse subsequent memory performance, i.e., updating of the object location in memory. The manuscript addresses an important topic: the relationship between memory processing and theta phase, and uses an interesting and innovative behavioral measure. However, I have several concerns regarding the analyses and the presentation of the results that impact the clarity of the conclusions.

1) I have concerns about the time course of the behavior and the analysis windows chosen. Table 1 shows that fixations last ~200-400 ms, which is consistent with a broad literature. However, many of the analyses use a window which surely encompasses more than the fixation of interest. For example, Figure 2B, the significant cluster of ITC occurs begins ~400 ms prior to the fixation. It is likely that this time window includes the fixation 2-back from the fixation of interest. Similarly, the post-fixation effect shown in Figure 2C likely includes the fixation of interest plus the following fixation. It seems that the cleanest approach would be to limit the analysis to the immediately preceding or following fixation. This may have been the approach taken, but it is unclear from the methods. Similarly, it seems likely that multiple fixations within a given ROI occur in succession (as illustrated in Figure 1B). It is not clear how this would have been controlled for in the analysis.

2) Because the presentation of the data are for the most part, fairly processed, it would be helpful to show an example of the raw LFP and theta phase-locking in an individual fixation. It is also very difficult to understand how many fixations were included in each of the analyses.

3) For the PAC analysis, it is unclear how the example of the representative electrode fits with the population result. The representative electrode showed increased PAC between theta (~5 Hz) and gamma (~140-170 Hz) for fixations to updated vs repeated objects. By contrast, the population effects showed significant PAC at 80-100 Hz. From Figure 5A, it does look like there may be a small increase in PAC in the 140-170 range, which did not reach significance. Is it possible that different PAC effects were observed at different electrode locations?

4) In the Discussion section, the authors report that "inspection of raw traces revealed that the observed differences in PAC resulted from nested oscillations, as opposed to the modulation in the amplitude of sharp waveforms." This is an interesting finding, and it would be helpful to provide examples of these raw traces.

5) During the Refresh phase, the subjects performed a recognition memory task, but I couldn't find these results reported. Importantly, it was not clear whether the subjects received feedback on the accuracy of their responses and how their responses on the Refresh phase correlated with accuracy in the final Test phase.

6) In the Materials and methods section, it is not clear whether a distinct background was used for each of the 16 objects or if the same background was used throughout a block. If a different background was used, was each background shown for a 5s familiarization period prior to the Study phase? It was also not clear whether the flashing of the fixation cross was in the location of the object. Was there one study trial per object/scene? Did all subjects perform all 8 blocks?*Reviewer #3:*The study "Hippocampal theta coordinates memory processing during visual exploration" presents behavioral data showing that viewing probability on a novel stimulus location on mismatch trials predicts later recognition memory, while looking behavior on match trials did not. The authors then selected fixations to mismatch locations and found that they were preceded by theta phase consistency in hippocampal recordings (Figure 2). Compared to repeat fixations, the fixations to updated locations resulted in stronger post-fixational ~4.5 Hz ITS (Figure 3). Additionally the study reports enhanced 5 to ~90Hz phase amplitude coupling for updating fixations and decreased ~5-140Hz phase amplitude coupling to a new object when the original location was forgotten as if it indicated an overriding of memory for the old object location association.

This is an interesting study, reporting findings in the human medial temporal lobe that extend prior studies in human and nonhuman primates showing that more consistent fixation aligned phases in the theta band relate to the updating and better remembering of newly fixated objects.

The study uses a well balanced design and applies state-of-the-art methods.

Beyond these strengths there are several aspects of the analysis and writing that deserve consideration.

1) A major concern with this study is the restricted frequency range analyzed which limits interpretation and understanding of memory updating related dynamics. The reader is presented with phase analysis results of the 1-10 Hz range and the 80-200 Hz range. This restriction does not allow to discern how overall (average) and beta and gamma activity relate to memory updating despite prior studies implicating these frequency ranges in related functions. What is the average time-locked average ERP activity around fixations (does that allows to see a reset like behavior)? Are there peri-fixation effects of ITC effects at beta and gamma activities that are related to updating and remembering fixated objects?

2) Were there power spectral peaks discernible at those frequencies at which phases are interpreted? If not, can this be discussed explicitly? What were the shapes of the power spectra? If power peaks at non-existent or at frequencies away from the max-ITC phase effects that would be important to know to constrain interpretation and guide future studies aimed at finding the neural basis of the ITC effects.

3) Figure 2 (legend and main text) insinuate that the analyzed fixations are all "retrieval-dependent". I think this is a misleading statement because it is not made sufficiently clear which fixations were used and only a small fraction may relate to retrieval. Were only Mismatch fixation used? Only from trials which were later 'remembered'? Including the first fixation?

If the behavior suggest that there are ~2 sec where refixations to the mismatch locations are predictive to the remembering, then fixations at other time windows are not linked to later remembering. This should be ideally made clear to the reader.

4) Related to the previous point, it would be appreciated to also see and report the pre- and post-fixational ITC for Match trials (e.g. in a figure supplement similar to Figure 2). If you would have a similar 5Hz ITC in those conditions then the interpretation of the pre-fixational mismatch ITC would be different.

5) It is unclear which "additional" features are possibly explored (given similar fixation durations) that is suggested by this sentence (subsection “Theta dependence of memory-guided eye movements”): " the number of fixations to the object was reduced on Mismatch relative to Match trials (paired t-test, t4 = -8.8, P = 0.0009, g = -0.6), indicating that subjects explored additional visual features during Mismatch".

6) With five subjects it seems necessary to report about the variability and consistency of the observed main findings more explicitly. How many subjects showed a positive memory effect for mismatch fixation trials (at which specific anatomical locations)? In how many subjects was there a predictive theta ITC? In how many subjects was there a post-fixational memory ITC effect?

7) How often were individual objects of the 16 objects shown (eight times?)? Were they shown the same numbers and durations? If no, then please test if this affects the theta ITC effect

8) Were objects shown in the same sequence in each of the eight blocks? If yes, please show that sequence effects are not explaining the memory predictive theta ITC.

9) Were the 16 objects shown at 16 random locations or were some shown at the same location? If the latter is the case, please provide test that the number of objects per location is not a confound.

[Editors’ note: further revisions were suggested prior to acceptance, as described below.]

Thank you for submitting your article "Hippocampal theta coordinates memory processing during visual exploration" for consideration by *eLife*. Your article has been reviewed by Laura Colgin as the Senior Editor, a Reviewing Editor, and three reviewers. The reviewers have opted to remain anonymous.

The reviewers have discussed the reviews with one another and the Reviewing Editor has drafted this decision to help you prepare a revised submission.

Summary:

This paper reports that eye movements are phase-locked to hippocampal theta rhythms in humans performing a spatial memory task. A major strength of the paper is its close link to theoretical predictions of distinct hippocampal theta phases being linked to novelty-related exploration/encoding and retrieval. The results provide quantitative data supporting this model in human subjects. However, there are several issues that require clarification and revisions that should be made to increase the impact of this work on a broad range of readers. Several features of the analyses need to be clarified for the results to be considered solid. In general, the authors should attempt to write up the results more clearly in a way that can be easily understood by the general readership of *eLife*.

Essential revisions:

1) Reviewers found the analysis that analyzed the phase angle of the resets to be compelling because it has the potential for increasing the impact of the work by linking to theoretical models from rodents. However, several concerns about this analysis were raised:

a) A concern was raised that it is possible that the 180 degree phase difference found between original and updated fixations is caused by the examination of a later relative time offset for fixations to updated positions. If this tricky analysis was understood correctly, it would be important to show that the apparent phase difference they report is truly a result of the resets occurring to different phases rather than just being a result of the analysis measuring phases at different time intervals in each of the two conditions.

b) The plot on the right panel of Figure 2E is very hard to read because the points are small and the blue and green colors are similar. This plot should be made more understandable, perhaps by showing separate histograms of the blue and green distributions as well as by performing a statistical test that is matched to the data (there is a description of some test in the text on this but it is hard to tell if this refers to the exact data in this figure?).

c) Regarding this result, an analysis at the individual electrode level was mentioned. Reviewers had a hard time understanding this analysis and thus it requires clarification. In general, it would be better if they could clarify what this effect means at the individual electrode level, perhaps by showing data from a representative electrode in both conditions. They should also better explain the statistical test in subsection “Theta dependence of memory-guided eye movements” where they mentioned that the effect is significant at 27 of 32 electrodes. It is presumed that they performed some sort of within-electrode, across-trials comparison (which test?). This test was described as "across electrodes" in the preceding sentence (subsection “Theta dependence of memory-guided eye movements”), which may be a mistake.

2) There is a lack of discussion of anatomical specificity of recordings. Specific points related to this are listed below:

a) A concern was raised about the claim that the findings are specific to the human hippocampus. The authors state that they included a mix of depth and grid electrodes and state the number of hippocampal contacts in each subject. These numbers were viewed as unlikely given that there are grid electrodes for one, and in some the intercontact spacing is 5 mm. Having 8 contacts in the hippocampus of a single subject at least with traditional epilepsy electrode implantations is high. Where and how did these 8 contacts come from – depth electrodes, most distal contact? The authors should further detail how these numbers were calculated in the subjects. Were depth electrodes implanted orthogonally? Figure 2A is really not helpful. For bipolar electrodes, how did the authors determine placement; was 1 or both of the contacts in the hippocampus? If indeed the electrode channels are from areas beyond the hippocampus, the authors should revise the claims of their paper accordingly. Furthermore, the number of contacts in Table 3 versus Table 4 does not match.

b) In its current form there is no fair consideration of where in the larger hippocampal formation the observed effects were measured when compared to other studies (in humans, nonhuman primates, and rodents). A discussion of this is necessary to allow interpreting the reported findings with prior signatures.

3) To a reader with knowledge about theta oscillations it is unclear how the reported theta oscillations in the human patients+ electrodes appear (the added waveform shape analysis does not help here). It seems pivotal to show example traces within a e.g. 1-20Hz bandpass signal. Showing these traces is necessary to allow comparison to findings in other studies/species.

4) With only 5 participants, it is possible that some of the results can be driven by a single subject or electrode(s). For example, a peak in theta (4-6 Hz) across the group can be reflected by a small number of electrodes or a few subjects. There seems to be a lot of variability in their group with respect to at least theta from the data that we can see. For specificity of the theta peak in each subject however, there is no way to tell as the paper stands currently. The above point about the larger anatomical concerns could potentially explain some of this variability.

5) What was the 60 second distraction task exactly? Were the subjects given a specific goal or task instruction?

6) A few concerns regarding presentation of statistical analyses were raised:

a) The sample sizes for which statistical analyses are done should be reported, in the text and in each associated figure legend panel. Currently, this is not the case.

b) With regards to Table 2, how was significance here determined?

[Editors’ note: further revisions were suggested prior to acceptance, as described below.]

Thank you for resubmitting your work entitled "Hippocampal theta coordinates memory processing during visual exploration" for further consideration by *eLife*. Your revised article has been evaluated by Laura Colgin (Senior Editor) and a Reviewing Editor.

The manuscript has been improved but there are some remaining issues that need to be addressed before acceptance, as outlined below:

*Reviewer #1:*The revised manuscript is considerably enhanced and now provides a rich set of important results (including several added control analysis that strengthen the authors main conclusions) and an extended and fair discussion.

The only aspect that I think deserves attention is the description of the PAC results and how they are interpreted:

a) Description of PAC results:

The authors report in subsection “Theta to gamma phase amplitude coupling predicts memory updating” that "[…] 25% of electrodes exhibited significant differences in PAC driven by associative novelty (i.e., differences between fixations to updated or repeated locations)[…]).

The critical question here, however, is not whether there are sign differences in more channels than expected by chance, but how many of them show enhanced PAC for updated locations?

It is confusing (and reflects overstating) to read e.g. subsection “Theta to gamma phase amplitude coupling predicts memory updating” that enhanced PAC when an object-location is forgotten is interpreted in the text as reflecting that memory is updated.

b) Interpretation:

The PAC results are interpreted as indicating not only that enhanced PAC is linked to forgetting (which seems backed up by the results), but that PAC is associated with memory updating. I do not see a result that suggests that a behavioral measure of updated memory is correlated with PAC. Hence this interpretation should not be made or some more explicit and direct result should be added to support that claim.

This affects the Abstract (which states "[…], but predicted memory updating") and it affects the heading of subsection “Theta to gamma phase amplitude coupling predicts memory updating”.

Reviewer #2:

The authors have done a nice job responding to my concern and I think the paper is appropriate for publication. I especially like the new Figure 3, which explains the results rather intuitively.

---

## [Author Response]

[Editors’ note: the authors resubmitted a revised version of the paper for consideration. What follows is the authors’ response to the first round of review.]

Reviewers all found the topic and experimental design compelling. However, all had significant concerns about the statistics and analyses that raised questions about whether the paper's conclusions were strongly supported by the results. The separate reviews are included below in their entirety, but major concerns that led to the rejection decision include:1) The time-window for analysis: It is unclear that the data only include the fixation of interest (and not any other saccade or fixation). Changing this window would dramatically decrease the available data for analysis to ~250 ms for each fixation.

The results in our manuscript necessarily include multiple behavioral events (eye movements) per time window, given the frequency of eye movements. In our revised manuscript, we provide an additional analysis that shows our findings are consistent when excluding individual fixations with temporally proximal saccades (clean windows ranging from 100 to 400 ms). These results are described in subsection “Theta dependence of memory-guided eye movements” of the text, and displayed in Figure 2—figure supplement 3.

2) Improper statistics: Analyses lacked a direct test of the interaction of interest (and instead concluded differential effects based on the finding that one result was significant and another was not).

We have carefully reviewed the manuscript for circumstances where we implied a statistical difference without explicitly testing for such differences. These circumstances were limited to two occasions, both in the statement regarding subsequent memory effects being specific to viewing of Updated locations on Mismatch trials (subsection “Theta phase consistency during updated-location viewing predicts subsequent memory”) and specificity of our PAC findings to the 4-6 Hz range (subsection “Theta to gamma phase amplitude coupling predicts memory updating”). We now provide statistical tests that directly compare the interaction between theta ITC and subsequent memory during fixations to Updated and Repeated object locations in support of this claim. Figure 3 now depicts subsequent memory effects for both viewing regions of interest, as well as the difference between these two conditions. This comparison is now reported in the main text in subsection “Rapid sequential fixations do not account for memory-specific theta effects”.

In addition, we now use a repeated measures ANOVA to compare PAC effects as a function of frequency band and memory condition (subsection “Theta to gamma phase amplitude coupling predicts memory updating”). In support of this claim, we now report tests showing that interactions between theta phase and gamma amplitude were greatest when using 4 to 6 Hz vs. higher or lower theta frequencies (subsection “Theta to gamma phase amplitude coupling predicts memory updating”).

We would also like to emphasize that the experimental design was not built off a factorial design, where it is straightforward to test for an interaction between conditions. While there are two experimental conditions of interest (Match and Mismatch trials), differences in memory processes only vary within the Mismatch condition. That is, while both novelty and retrieval can guide fixations in the Mismatch condition, comparable viewing states are not present in the Match condition. As a result, an interaction analysis is not practical for the majority of the reported analyses. The comparison to Match trial acts as an important control, demonstrating, for example, that differences in theta phase are not primarily driven by fixating on the presented object.

3) The PAC results were viewed as a weak part of the paper. Concerns were raised that PAC effects may have resulted from changes in power or waveform asymmetry and that analyses were not corrected for multiple comparisons.Differences in the gamma frequency observed (very fast) compared to previous results in rodents (e.g., Colgin et al.) made the discussion of the results as related to slow and fast gamma strange.

We agree with the reviewers that we did not provide strong evidence that our PAC results were not driven by changes in underlying properties of the theta waveform. We now include additional control analyses (subsection “Theta to gamma phase amplitude coupling predicts memory updating”) that demonstrate that while measures of PAC were influenced by theta sharpness and power, they do not account for the observed differences in PAC related to memory. In addition, we have refocused our discussion to similar PAC findings in humans, where appropriate.

4) There was a lack of demonstration of peaks in the spectra for the frequency ranges that were chosen for analysis to justify why these particular frequencies were selected. It is possible that similar results observed at other frequencies and that drawing conclusions about specific oscillatory patterns are unwarranted.

We now demonstrate clear peaks in the power spectra from 4 to 6 Hz across patients and conditions (Figure 2). We note that our focus on theta was driven by previous findings in nonhuman primates (e.g. Jutras et al., 2013) relating hippocampal theta to memory encoding. To provide some evidence regarding the specificity of these results with regard to higher frequency oscillations, we provide an additional exploratory analysis that looks at effects from 1 to 50 Hz, which fails to identify robust effects outside the theta/alpha range (Figure 4—figure supplement 2).

5) Nothing is shown to describe inter-subject variability.

We now characterize the degree to which ITC (and PAC) effects vary across individual subjects. While greater anatomical description of these effects (e.g., comparing across hippocampal subfields or along the anterior-posterior axis) would be ideal, we do not have sufficient coverage across this group of patients to examine variability within different regions of the hippocampus. This information can be found in Table 2, Table 3, Supplementary file 2 and Supplementary file 3.

6) Pre- and post- fixation results were not shown for Match trials (i.e., similar to Figure 2).

We now display the pre- and post-fixation results comparing fixations to individual ROIs on Match and Mismatch trials. Graphical depictions of these comparisons were excluded from the initial manuscript to focus the message to significant findings. The results of this analysis are now graphically depicted in Figure 4—figure supplement 1.

Reviewer #1:[…]- The most substantial concern I see with the paper is that many of the paper's claims hinge on a comparison of two statistical tests, with the authors making much of the fact that one comparison is significant and that another is not. Instead the paper needs to directly test whether one effect is larger than the other, such as by testing the interaction with an ANOVA or some other way. This is a problem in many sections of the paper and substantially undermines the specificity of key claims. The authors should review all their claims and ensure that result is supported by a specific statistical test/interaction, rather than just relying on an effect being significant in one condition and not in another statistically.

We thank the reviewer for bringing this point to our attention. We have carefully reviewed the manuscript and included additional statistical tests, where necessary. See our response to general comment 2 for more details.

- I was disappointed that the paper only measured the magnitude of phase locking, while seemingly ignoring the key issue of what specific phase the locking occurred to. The models that they refer to from Hasselmo and Colgin have strong hypotheses related to specific theta phases and the recorded data would seem to measure this data. But unfortunately, the reported data analyses ignore testing which theta phase the locking occurs to, instead focusing only the magnitude of locking. Without this information, the paper doesn't really provide strong support for those theoretical models.

We agree with the reviewer that strong support for these theoretical models would show phase-specific effects. To test these theories, we identified the preferred phase angles at the onset of each cluster related to novelty and retrieval processes (Figure 2E, left). As shown in the right panel of Figure 2E, we found that phase-locking associated with retrieval began near the peak of theta, whereas novelty-related effects were timed to the trough of theta. A more detailed description of these results can be found in subsection “Theta dependence of memory-guided eye movements”. We thank the reviewer for suggesting this additional analysis, as it directly ties our findings to these theories of hippocampal theta function.

- It is notable that the latencies of the effects are close to 300ms post saccade. Are the effects related to P300s?

We performed additional analysis of ERPs to individual fixations, to examine the possibility that (1) phase locking at different frequencies and (2) ERP components are related to the observed differences in ITC across conditions.

**Author response image 1. respfig1:** Mean event related potentials (ERPs) across all patients and electrodes. Shaded regions indicate SEM.

- The PAC analyses are not compelling because they do not rule out the possibility that the effects are driven by changes in power. Based on the data that they show, I think it remains possible that their PAC results are a direct result of changes in theta power or locking without any specific PAC changes. To substantiate their claim that their findings demonstrate a specific role for PAC in memory, the authors need to do much more to specifically show that their effects are caused by interactions between the timing of gamma and theta rather than power or phase changes in just one of the signals. They also should rule out whether the PAC changes could be related to waveform asymmetry.

We agree with the reviewer that it is important to formally test whether interactions between theta and gamma oscillations are responsible for the observed differences in PAC, as opposed to changes in power or underlying theta waveforms. The current version of the manuscript now includes control analyses to rule out the possibility that PAC changes resulted from differences in (1) theta power, (2) theta phase-locking, (3) waveform sharpness, or (4) waveform asymmetry (these results are described in the new subsection “Memory-related changes in PAC are unrelated to theta waveform properties”).

For each significant memory-related PAC effect (i.e., those reported in Figure 5A and 5B), we examined whether underlying properties of the theta waveform predicted concurrent changes in gamma amplitude. While we did find significant (but modest) relationships between multiple theta waveform properties at individual electrodes (and at the group level related to theta power; see Supplementary file 2 and Supplementary file 3), we performed a regression analyses to remove any variance in PAC related to these factors. The results of these analyses, which show consistent memory-related modulates in PAC, are depicted in Figure 6.

In our first analysis, we quantified changes in the LFP evoked by individual fixations to each ROI. Grand average (across electrodes and patients) ERPs are depicted in the following figure. At the group level, we observed modest differences in event related potentials 600 ms after fixations to original vs. updated locations on Mismatch trials (Author response image 1). In general, ERP effects were heterogenous across electrodes and subjects, leading to these results. These results argue against the idea that observed theta effects are driven by evoked responses, such as the hippocampal P300.

- I had a hard time following the control analyses in the third paragraph of subsection “Theta dependence of memory-guided eye movements”. Can the text here be reworded so the logic is clearer?

We have revised the manuscript in subsection “Theta dependence of memory-guided eye movements*”* to clarify these control analyses.

- I was surprised to see high-gamma effects at frequencies as high as 130-150 Hz. Were these results corrected for multiple comparisons? In the Discussion section this pattern is compared to the fast and slow gamma oscillations seen in rodents by Colgin et al. However, this signal is so substantially faster in frequency than the signals seen in rodents that this comparison seems far-fetched.

These analyses were corrected for multiple comparisons (accounting for frequencies and angles for phase), but not for the three different task-based comparisons.

We have modified our Discussion section such that we do not directly make inferences regarding the frequency of gamma oscillations. However, we do believe that there is frequency specificity in the observed PAC findings, which supports the general notion that the speed of gamma-band activity may reflect the function of distinct neural circuits/cognitive functions in humans, as they do in other model systems.

- It seems that most of the paper's statistics are performed across subjects (based on the degrees of freedom). This is a useful, conservative approach, which they should better explain this in the results. They used a different approach in subsection “Theta to gamma phase amplitude coupling predicts memory updating”, when they report percentages of hippocampal electrodes that show each effect, which they should explain and justify.

Our electrode-level analyses were meant to convey the degree to which these effects were observable at a given recording site, as opposed to the typical group-level analysis (treating subjects as random effects) used in neuroimaging studies. In response to other reviewers’ concerns, we now uniformly report electrode and subject-level statistics across all analyses (notably, we refer to this statistical decision in subsection “Theta dependence of memory-guided eye movements”, where we refer to new subject and electrode level ITC effects reported in Table 3).

Reviewer #2:[…]1) I have concerns about the time course of the behavior and the analysis windows chosen. Table 1 shows that fixations last ~200-400 ms, which is consistent with a broad literature. However, many of the analyses use a window which surely encompasses more than the fixation of interest. For example, Figure 2B, the significant cluster of ITC occurs begins ~400 ms prior to the fixation. It is likely that this time window includes the fixation 2-back from the fixation of interest. Similarly, the post-fixation effect shown in Figure 2C likely includes the fixation of interest plus the following fixation. It seems that the cleanest approach would be to limit the analysis to the immediately preceding or following fixation. This may have been the approach taken, but it is unclear from the methods. Similarly, it seems likely that multiple fixations within a given ROI occur in succession (as illustrated in Figure 1B). It is not clear how this would have been controlled for in the analysis.

The reviewer is correct that the time-window of interest will necessarily include extra fixation events potentially before and after the fixation indicated at time zero. As such, any analysis of these time periods requires a trade-off between statistical power (including more trials) and potential confounds that can influence theta (additional events occurring in the time window of interest). Given our interest in characterizing theta power and phase, we necessarily require multiple cycles of a theta to obtain reliable estimates, on the order of seconds for lower frequencies. As a result, only examining ‘fixation free’ epochs would drastically reduce the number of observations and statistical power to detect any effects.

To provide evidence that our findings are not driven by sequential sampling behaviors (e.g., a sequence of fixations from the updated to the original object location in the Mismatch condition), we have repeated the main phase-locking analyses with different ‘clean windows’ where no additional fixations occur in either a pre- or post-fixation time period. These analyses replicated the main findings in the manuscript and are reported in subsection “Theta dependence of memory-guided eye movements” and depicted in Figure 2—figure supplement 3. We additionally provide a description of the number of fixations included in each of these analyses in Supplementary file 1.

2) Because the presentation of the data are for the most part, fairly processed, it would be helpful to show an example of the raw LFP and theta phase locking in an individual fixation. It is also very difficult to understand how many fixations were included in each of the analyses.

As mentioned above, we now provide more detailed information regarding the number of fixations that are included in each analysis. The raw hippocampal LFP with individual fixation events for a single Mismatch trial are provided in Figure 1.

3) For the PAC analysis, it is unclear how the example of the representative electrode fits with the population result. The representative electrode showed increased PAC between theta (~5 Hz) and gamma (~140-170 Hz) for fixations to updated vs repeated objects. By contrast, the population effects showed significant PAC at 80-100 Hz. From Figure 5A, it does look like there may be a small increase in PAC in the 140-170 range, which did not reach significance. Is it possible that different PAC effects were observed at different electrode locations?

As noted in subsection “Reset of hippocampal oscillations during memory-guided eye movements is specific to theta”, we observed a great degree of heterogeneity across individual electrodes regarding the frequencies which exhibit PAC. As a result, not every electrode showed a result that typified the group-level analysis. To avoid potential confusion, we have selected a different example electrode for Figure 4, which is more consistent with group-level results.

4) In the Discussion section, the authors report that "inspection of raw traces revealed that the observed differences in PAC resulted from nested oscillations, as opposed to the modulation in the amplitude of sharp waveforms." This is an interesting finding, and it would be helpful to provide examples of these raw traces.

We now include analysis of theta waveform properties (see our response to issue 3 raised by reviewer 1) and have amended our Discussion section to refer to these analyses directly.

5) During the Refresh phase, the subjects performed a recognition memory task, but I couldn't find these results reported. Importantly, it was not clear whether the subjects received feedback on the accuracy of their responses and how their responses on the Refresh phase correlated with accuracy in the final Test phase.

We now include behavioral performance during the Refresh phase (the same/different judgment) and differences in initial and final responses in the behavioral results subsection “Direct brain recordings linked to memory-driven eye movements”. Subjects were not provided feedback regarding the accuracy of their response (Materials and methods section), and performance was highly correlated across phases of the task.

6) In the Materials and methods section, it is not clear whether a distinct background was used for each of the 16 objects or if the same background was used throughout a block. If a different background was used, was each background shown for a 5s familiarization period prior to the Study phase? It was also not clear whether the flashing of the fixation cross was in the location of the object. Was there one study trial per object/scene? Did all subjects perform all 8 blocks?

A unique background was used for each set of 16 objects (each block), and each block used trial-unique objects. All subjects performed all 8 blocks of the task. We now clarify these details in the Materials and methods section and Results section.

Reviewer #3:[…]Beyond these strengths there are several aspects of the analysis and writing that deserve consideration.1) A major concern with this study is the restricted frequency range analyzed which limits interpretation and understanding of memory updating related dynamics. The reader is presented with phase analysis results of the 1-10 Hz range and the 80-200 Hz range. This restriction does not allow to discern how overall (average) and beta and gamma activity relate to memory updating despite prior studies implicating these frequency ranges in related functions. What is the average time-locked average ERP activity around fixations (does that allows to see a reset like behavior)? Are there peri-fixation effects of ITC effects at beta and gamma activities that are related to updating and remembering fixated objects?2) Were there power spectral peaks discernible at those frequencies at which phases are interpreted? If not, can this be discussed explicitly? What were the shapes of the power spectra? If power peaks at non-existent or at frequencies away from the max-ITC phase effects that would be important to know to constrain interpretation and guide future studies aimed at finding the neural basis of the ITC effects.

We now display power spectra from hippocampal contacts (Figure 2B) which reveal clear spectral peaks in the 4 to 6 Hz range. We also fit power spectra using a combination of periodic and aperiodic components, and assessed statistical significance of spectral peaks (using an F-test) at each electrode in each experimental condition (Table 2). At the group level, we found that more than half of the electrodes exhibited statistically significant oscillations in this frequency range.

3) Figure 2 (legend and main text) insinuate that the analyzed fixations are all "retrieval-dependent". I think this is a misleading statement because it is not made sufficiently clear which fixations were used and only a small fraction may relate to retrieval. Were only Mismatch fixation used? Only from trials which were later 'remembered'? Including the first fixation?If the behavior suggest that there are ~2 sec where refixations to the mismatch locations are predictive to the remembering, then fixations at other time windows are not linked to later remembering. This should be ideally made clear to the reader.

In Figure 2 and in the main text, we refer to fixations to the original object-location on Mismatch trials (when there is no object present at this location) as retrieval-dependent. As shown in Figure 1D, indicated in blue, fixations to the original object location almost exclusively occurred later in the fixation sequence, following viewing of the presented object. We have clarified in the Results section what we mean by retrieval-dependent.

4) Related to the previous point, it would be appreciated to also see and report the pre- and post-fixational ITC for Match trials (e.g. in a figure supplement similar to Figure 2). If you would have a similar 5Hz ITC in those conditions then the interpretation of the pre-fixational mismatch ITC would be different.

We now include these results in Figure 2—figure supplement 3; as reported in the main text, there is no significant difference in ITC between these conditions.

5) It is unclear which "additional" features are possibly explored (given similar fixation durations) that is suggested by this sentence (subsection “Theta dependence of memory-guided eye movements”) : " the number of fixations to the object was reduced on Mismatch relative to Match trials (paired t-test, t4 = -8.8, P = 0.0009, g = -0.6), indicating that subjects explored additional visual features during Mismatch".

We have clarified that we were referring to the background scene.

6) With five subjects it seems necessary to report about the variability and consistency of the observed main findings more explicitly. How many subjects showed a positive memory effect for mismatch fixation trials (at which specific anatomical locations)? In how many subjects was there a predictive theta ITC? In how many subjects was there a post-fixational memory ITC effect?

Our focus on the theta and gamma frequencies was well motivated by prior work in nonhuman primates and humans. Nonetheless, we have performed an additional ERP analysis to examine the extent to which phaselocking may occur at additional frequencies (see Author response image 1. While we did not observe reliable differences in ERPs across patients and electrodes, we examined changes in spectral power of these ERPs at the electrode levels to identify differences in phase alignments across a broader range of frequencies. As shown in Figure 4—figure supplement 2, increases in pre- vs. post-fixation onset ERP power are observed in the high theta/alpha range (8 to 13 Hz) following fixations to objects during both match and mismatch conditions. These results highlight the specificity of our findings to the theta range and provide additional evidence regarding optimal stimulus encoding via resetting of ongoing theta oscillations. These findings are now reported in the subsection “Reset of hippocampal oscillations during memory-guided eye movements is specific to theta”.

Variability across subjects and electrodes is now described in subsection “Theta dependence of memory-guided eye movements” and Table 2.

7) How often were individual objects of the 16 objects shown (eight times?)? Were they shown the same numbers and durations? If no, then please test if this affects the theta ITC effect

All objects used in the study were trial unique. That is to say, each object appeared once (during study, refresh, and test phases) on the background scene that was used for a given block.

8) Were objects shown in the same sequence in each of the eight blocks? If yes, please show that sequence effects are not explaining the memory predictive theta ITC.

Objects were not repeated across blocks. As a result, there is no possibility for sequential learning across blocks.

9) Were the 16 objects shown at 16 random locations or were some shown at the same location? If the latter is the case, please provide test that the number of objects per location is not a confound.

Objects were shown in unique locations.

[Editors’ note: what follows is the authors’ response to the second round of review.]

Essential revisions:1) Reviewers found the analysis that analyzed the phase angle of the resets to be compelling because it has the potential for increasing the impact of the work by linking to theoretical models from rodents. However, several concerns about this analysis were raised:a) A concern was raised that it is possible that the 180 degree phase difference found between original and updated fixations is caused by the examination of a later relative time offset for fixations to updated positions. If this tricky analysis was understood correctly, it would be important to show that the apparent phase difference they report is truly a result of the resets occurring to different phases rather than just being a result of the analysis measuring phases at different time intervals in each of the two conditions.

We agree with the reviewers that it is important to demonstrate that differences in theta phase related to retrieval and associative novelty do not directly result from differences in the timing of these two effects, relative to fixation onsets. We now present a comparison of phase distributions considering both the viewing region of interest (i.e., updated or original object-location) and time interval (i.e., pre or post fixation onset) to rule out this possibility (see Figure 3). We find two pieces of evidence that support our conclusions. First, in the interval preceding fixations to updated object-locations, there is no evidence that the phase distribution is different from a uniform distribution (*Z* = 1.04, *P* = 0.36, Rayleigh test). That is, theta phase is not consistently aligned to a specific angle in this condition, prior to execution of the saccade. Second, following fixations to the original object location, we found theta phase distributions concentrated around the trough of the oscillation (mean = 2.9 rad [2.5-3.4 95% CI]). These phase angles were not distinguishable from those following fixations to the updated object-location (*F*(62) = 0.77, *P* = 0.38). Thus, theta phase following fixations on Mismatch trials are generally aligned to the trough; however, alignment to the peak of theta only occurs when we believe retrieval processes are necessary (preceding fixations to the original object-location). We have added a statement clarifying this point in the Results section.

b) The plot on the right panel of Figure 2E is very hard to read because the points are small and the blue and green colors are similar. This plot should be made more understandable, perhaps by showing separate histograms of the blue and green distributions as well as by performing a statistical test that is matched to the data (there is a description of some test in the text on this but it is hard to tell if this refers to the exact data in this figure?).

We agree that the results presented in Figure 2E were difficult to read. To more prominently display these findings, we have added an additional figure (Figure 3) that presents these results both in the form of polar histograms and in addition to the previous format (which we believe clearly communicates the peak/trough relationship to reader unfamiliar with polar plots). We have also rewritten the text in the Results section to precisely follow what is depicted in the figure.

c) Regarding this result, an analysis at the individual electrode level was mentioned. Reviewers had a hard time understanding this analysis and thus it requires clarification. In general, it would be better if they could clarify what this effect means at the individual electrode level, perhaps by showing data from a representative electrode in both conditions. They should also better explain the statistical test in subsection “Theta dependence of memory-guided eye movements” where they mentioned that the effect is significant at 27 of 32 electrodes. It is presumed that they performed some sort of within-electrode, across-trials comparison (which test?). This test was described as "across electrodes" in the preceding sentence (subsection “Theta dependence of memory-guided eye movements”), which may be a mistake.

In order to make this analysis more intuitive to readers, we now describe the procedure in a descriptive manner from measuring theta phase relative to specific fixations (Figure 3B), to comparing differences at the level of individual electrodes (Figure 3C), and finally across all electrodes (Figure 3D). We also thank the reviewers for catching an error introduced during editing, which has been corrected.

2) There is a lack of discussion of anatomical specificity of recordings. Specific points related to this are listed below:a) A concern was raised about the claim that the findings are specific to the human hippocampus. The authors state that they included a mix of depth and grid electrodes and state the number of hippocampal contacts in each subject. These numbers were viewed as unlikely given that there are grid electrodes for one, and in some the intercontact spacing is 5 mm. Having 8 contacts in the hippocampus of a single subject at least with traditional epilepsy electrode implantations is high. Where and how did these 8 contacts come from – depth electrodes, most distal contact? The authors should further detail how these numbers were calculated in the subjects. Were depth electrodes implanted orthogonally? Figure 2A is really not helpful. For bipolar electrodes, how did the authors determine placement; was 1 or both of the contacts in the hippocampus? If indeed the electrode channels are from areas beyond the hippocampus, the authors should revise the claims of their paper accordingly. Furthermore, the number of contacts in table 3 versus 4 does not match.

We agree with the reviewers that the number of contacts is high for typical electrode implantations with lateral trajectories targeting the hippocampus. We now clarify two points in the methods: (1) while patients were implanted with subdural grids, all analyses focused on depths targeting hippocampus, and (2) following bipolar referencing, all pairs with at least one contact within grey matter or proximal tissue were analyzed. This led to our inclusion of 31 contacts (yielding 32 bipolar pairs, accounting for the discrepancy between Table 3 and Table 4). In addition, we have updated Figure 2A to clarify the position of contact locations within each patient within the hippocampus. Even though recordings were lateralized, increased hippocampal coverage resulted from multiple hippocampal depth electrodes per patient.

b) In its current form there is no fair consideration of where in the larger hippocampal formation the observed effects were measured when compared to other studies (in humans, nonhuman primates, and rodents). A discussion of this is necessary to allow interpreting the reported findings with prior signatures.

In our previous iteration of the manuscript, we omitted discussion of anatomical localization within the hippocampus as our coverage was limited to the head and body of the hippocampus. We have expanded our discussion to include comparisons of our results to other studies, with specific focus on understanding functional organization across the long axis of the hippocampus. We also emphasize limitations of our dataset with regards to the anatomical specificity of our findings, as we did not obtain recordings in posterior hippocampus or resolve difference within hippocampal subfields.

3) To a reader with knowledge about theta oscillations it is unclear how the reported theta oscillations in the human patients+ electrodes appear (the added waveform shape analysis does not help here). It seems pivotal to show example traces within a e.g. 1-20Hz bandpass signal. Showing these traces is necessary to allow comparison to findings in other studies/species.

We now provide additional examples of raw traces, both filtered below 20 Hz and within the predominant theta frequency (4-6 Hz) in Figure 3 of the main text. To provide additional examples that may highlight anatomical variability, additional traces are shown in Figure 2—figure supplement 1.

4) With only 5 participants, it is possible that some of the results can be driven by a single subject or electrode(s). For example, a peak in theta (4-6 Hz) across the group can be reflected by a small number of electrodes or a few subjects. There seems to be a lot of variability in their group with respect to at least theta from the data that we can see. For specificity of the theta peak in each subject however, there is no way to tell as the paper stands currently. The above point about the larger anatomical concerns could potentially explain some of this variability.

The fact that electrode and subject effects can drive grand averages is precisely the reason we evaluated the presence of 4-6 Hz oscillations at the individual electrode level. As reported in Table 2, nearly 80% of electrodes exhibited statistically significant oscillations at this frequency irrespective of condition, except for after fixations to the original object-location (which did not involve viewing of a presented stimulus) in which only 50% of bipolar pairs exhibited significant oscillatory activity. We now present power spectra for individual electrodes in Figure 2—figure supplement 1, which demonstrates consistency of this effect.

5) What was the 60 second distraction task exactly? Were the subjects given a specific goal or task instruction?

During the distraction task, subjects performed free-viewing of unrelated visual scenes (cats). The stimulus pool and task are now described in the Materials and methods section.

6) A few concerns regarding presentation of statistical analyses were raised:a) The sample sizes for which statistical analyses are done should be reported, in the text and in each associated figure legend panel. Currently, this is not the case.

We now specify sample sizes for all tests and summary measures.

b) With regards to Table 2, how was significance here determined?

To determine whether statistically significant oscillations were present at each electrode, we modeled the power spectra for each period of interest (i.e., each epoch around a given fixation) as a mixture of aperiodic (modeled as 1/f) and periodic (modeled as gaussians) components. After model fitting, we performed a test of equal variance (across fixations) between the raw power spectra and the power spectra after removing modeled oscillations, yielding an *F*-statistic. If oscillations were present in the data, one would expect unequal variances across these two distributions. Statistical significance for each electrode was assessed using a permutation procedure (n = 10,000), randomizing assignment of psd-type (i.e., raw or aperiodic) across fixations. Statistical tests at the subject and group level were performed using binomial tests, comparing the proportion of significant electrodes to chance (established as 0.05 in the permutation procedure), with an α of 0.05, Bonferroni corrected for multiple comparisons across conditions and time periods (i.e., pre- post-fixation).

This description of model fitting is explained in the subsection “Spectral decomposition”. We have added description of the statistical procedure to the subsection “Statistical analyses”.

[Editors' note: further revisions were suggested prior to acceptance, as described below.]

Reviewer #1:The revised manuscript is considerably enhanced and now provides a rich set of important results (including several added control analysis that strengthen the authors main conclusions) and an extended and fair discussion.The only aspect that I think deserves attention is the description of the PAC results and how they are interpreted:a) Description of PAC results:The authors report in subsection “Theta to gamma phase amplitude coupling predicts memory updating” that "[…] 25% of electrodes exhibited significant differences in PAC driven by associative novelty (i.e., differences between fixations to updated or repeated locations)[…]).The critical question here, however, is not whether there are sign differences in more channels than expected by chance, but how many of them show enhanced PAC for updated locations?It is confusing (and reflects overstating) to read e.g. subsection “Theta to gamma phase amplitude coupling predicts memory updating” that enhanced PAC when an object-location is forgotten is interpreted in the text as reflecting that memory is updated.

In this analysis, we were primarily concerned with whether PAC was modulated by associative novelty, rather than focusing on absolute levels of PAC within a given condition (e.g., comparing PAC during fixations to updated locations against a reasonable null, such as surrogates constructed by permuting phase information across fixations). We agree that testing for differences in PAC during the viewing of updated locations would provide additional evidence that PAC is related to memory updating. We now report the suggested analysis (subsection “Theta to gamma phase amplitude coupling predicts memory updating”), comparing PAC following fixations to updated object-locations to phase-permuted null distributions. We found that 16% of hippocampal electrodes (significantly more than expected by chance, *P* = 0.02, binomial test) exhibited increased theta to gamma PAC. Based on these results and our analysis of forgetting-related changes in PAC, we do not believe we are overstating our findings.

b) Interpretation:The PAC results are interpreted as indicating not only that enhanced PAC is linked to forgetting (which seems backed up by the results), but that PAC is associated with memory updating. I do not see a result that suggests that a behavioral measure of updated memory is correlated with PAC. Hence this interpretation should not be made or some more explicit and direct result should be added to support that claim.This affects the Abstract (which states "[…], but predicted memory updating") and it affects the heading of subsection “Theta to gamma phase amplitude coupling predicts memory updating”.

Our claim that increased PAC is associated with memory updating is based on both (1) greater PAC during fixations to updated vs. repeated object-locations and (2) greater PAC during fixations to updated object-locations when the original location is forgotten (vs. remembered). This inference does rely on behavior on the task. When subjects forgot the original object-location on mismatch trials, they reported the updated object-location on 73% of trials. That is to say, interference from the refresh phase disrupted memory for the original location. We now clarify this point when presenting the results (subsection “Theta to gamma phase amplitude coupling predicts memory updating”). In our discussion of these results (Discussion section), we now emphasize that forgetting on the task was primarily due to updating of the object location. This subtlety is important when considering the role of hippocampal PAC in memory formation, which is commonly evaluated during initial encoding.